# Genome assemblies of 11 bamboo species highlight diversification induced by dynamic subgenome dominance

Peng-Fei Ma [1,10], Yun-Long Liu [1,10], Cen Guo[1,2,10], Guihua Jin[1,10], Zhen-Hua Guo[1,10], Ling Mao[1,3], Yi-Zhou Yang [1,3], Liang-Zhong Niu[1], Yu-Jiao Wang[1], Lynn G. Clark[4], Elizabeth A. Kellogg [5], Zu-Chang Xu [1,3], Xia-Ying Ye[1], Jing-Xia Liu[1], Meng-Yuan Zhou [1], Yan Luo [2], Yang Yang[1], Douglas E. Soltis [6,7], Jeffrey L. Bennetzen [8], Pamela S. Soltis [6] & De-Zhu Li [1,3,9] ✉

Polyploidy (genome duplication) is a pivotal force in evolution. However, the interactions between parental genomes in a polyploid nucleus, frequently involving subgenome dominance, are poorly understood. Here we showcase analyses of a bamboo system (Poaceae: Bambusoideae) comprising a series of lineages from diploid (herbaceous) to tetraploid and hexaploid (woody), with 11 chromosome-level de novo genome assemblies and 476 transcriptome samples. We find that woody bamboo subgenomes exhibit stunning karyotype stability, with parallel subgenome dominance in the two tetraploid clades and a gradual shift of dominance in the hexaploid clade. Allopolyploidization and subgenome dominance have shaped the evolution of tree-like lignified culms, rapid growth and synchronous flowering characteristic of woody bamboos as large grasses. Our work provides insights into genome dominance in a remarkable polyploid system, including its dependence on genomic context and its ability to switch which subgenomes are dominant over evolutionary time.

As a main driving force in evolution, polyploidy is ubiquitous across the green plant tree of life[1,2]. The resulting genic redundancy is a source of genetic innovation[2,3]. However, following genome doubling, the component subgenomes must cooperate to mediate potential incompatibilities of gene dosage, regulatory controls and transposable element (TE) activity[4,5]. Often, the evolution of subgenome dominance could be a solution and contributes substantially to species adaptation and diversification[4,6,7], although dominance may be minor or nonexistent in polyploids such as oats and teff[8,9]. Furthermore, most insights about dominance are limited to recently (a few million years ago (Ma)) formed polyploid crops (for example, wheat, cotton and brassicas) and their wild relatives that have not undergone extensive species

[1]Germplasm Bank of Wild Species & Yunnan Key Laboratory of Crop Wild Relatives Omics, Kunming Institute of Botany, Chinese Academy of Sciences, Kunming, Yunnan, China. [2]Center for Integrative Conservation & Yunnan Key Laboratory for the Conservation of Tropical Rainforests and Asian Elephants, Xishuangbanna Tropical Botanical Garden, Chinese Academy of Sciences, Menglun, Mengla, Yunnan, China. [3]Kunming College of Life Science, University of Chinese Academy of Sciences, Kunming, Yunnan, China. [4]Department of Ecology, Evolution, and Organismal Biology, Iowa State University, 345 Bessey, Ames, IA, USA. [5]Donald Danforth Plant Science Center, St. Louis, MO, USA. [6]Florida Museum of Natural History, University of Florida, Gainesville, FL, USA. [7]Department of Biology, University of Florida, Gainesville, FL, USA. [8]Department of Genetics, University of Georgia, Athens, GA, USA. [9]Key Laboratory for Plant Diversity and Biogeography in East Asia, Kunming Institute of Botany, Chinese Academy of Sciences, Kunming, Yunnan, China. [10]These authors contributed equally: P.-F. Ma, Y.-L. Liu, C. Guo, G. Jin, Z.-H. Guo. ✉e-mail: dzl@mail.kib.ac.cn

diversification[6,10,11]. Hence, we have limited understanding of how subgenomes differentially evolved in ancient polyploids that have founded major lineages with extensive species diversification.

Bamboos comprise the monophyletic Bambusoideae in Poaceae with a minor herbaceous, essentially diploid clade (126 species) and three major polyploid woody clades (1,576 species)[12]. The woody bamboos (WBs) exhibit distinctive biological traits, including highly lignified culms, rapid growth (up to 114.5 cm daily) and synchronous, usually monocarpic, flowering (~30–60 years)[13,14]. They are also of great cultural, ecological and economic importance in many parts of the Americas, Africa and Asia; the gross output of the bamboo industry in China alone reached ~$46 billion in 2020 (ref. 15).

Previous studies of bamboos identified two independent tetraploidizations followed by a hexaploidization event, all around 20 Ma in WBs, involving unresolved hypotheses with three[16], four[17] or five extinct diploid lineages[18]. Generally constant chromosome numbers have been reported for WBs (for example, $2n = (40)46–48$ for tetraploids and $2n = 70–72$ for hexaploids)[19,20], suggesting that the component subgenomes have likely remained unreshuffled. Hence, bamboos provide an ideal system for studying the evolution of subgenome dominance in plants of ancient polyploid origin.

## Results

### Sequencing of 11 bamboo genomes

As the third largest grass subfamily, the Bambusoideae show great diversity in species and morphology[12,19,21] (Fig. 1a and Extended Data Fig. 1a–k). To cover different ploidal levels and phylogenetic diversity, we selected 11 representative species for genome sequencing: two herbaceous bamboos (HBs, $2x$, *Olyra latifolia* and *Raddia guianensis*) and nine WBs of three clades: temperate (TWBs, $4x$, *Ampelocalamus luodianensis*, *Hsuehochloa calcarea* and *Phyllostachys edulis*), neotropical (NWBs, $4x$, *Rhipidocladum racemiflorum*, *Otatea glauca* and *Guadua angustifolia*) and paleotropical (PWBs, $6x$, *Melocanna baccifera*, *Bonia amplexicaulis* and *Dendrocalamus sinicus*) (Fig. 1b and Extended Data Table 1). Among these, *D. sinicus* is the largest known bamboo in the world, in sharp contrast to the herbaceous *Ra. guianensis* (Fig. 1c,d).

Combining coverage from an average of 124.5x Nanopore long reads (Supplementary Table 1) and 80.4x short reads, the 11 genomes were assembled de novo and polished into 114 to 3,619 contigs, with an average and maximum N50 of 5.3 Mb and 17.5 Mb, respectively. Using chromatin conformation capture (Hi-C) sequencing, an average of 94.1% of the sequences from the 11 genomes were anchored and assembled consistently into 11, 24 and 35 pseudo-chromosomes in diploid, tetraploid and hexaploid species (Fig. 1b and Supplementary Fig. 1), respectively; *G. angustifolia* was the single exception, with 23 pseudo-chromosomes as reported[19,20]. Moreover, chromosome-level synteny with a 1:2:3 pattern between the rice genome, often used as a reference in grasses[22], and the diploid, tetraploid and hexaploid bamboo genomes, respectively, was recovered (Supplementary Fig. 2), consistent with the expected ploidal levels from chromosome counts.

The high contiguity and completeness of the assemblies were supported by evidence from short-read mapping (an average of 98.9% ratio and all above 95.0%) (Supplementary Table 2) and LTR Assembly Index (LAI) (all assemblies qualified at the reference level or above with

LAI ≥ 10)[23] (Extended Data Fig. 1m). We annotated an average of 29,343, 47,444 and 51,989 protein-coding genes for diploid, tetraploid and hexaploid genomes (Supplementary Table 3), respectively, supported by 93.2% to 99.0% (average 96.4%) Benchmarking Universal Single-Copy Orthologue (BUSCO)[24] completeness (Extended Data Fig. 1l). High accurately assembled genes (AG) scores were also obtained by Mabs[25] with consistent sequencing coverage for single- and multicopy genes (Extended Data Fig. 1n and Supplementary Fig. 3). Together, these results indicated the high quality of all assembled genomes.

Genome sizes ranged from an average of 625.9 Mb in diploid to 1,628.3 Mb in tetraploid to 1,122.4 Mb in hexaploid bamboos, with 62.4%, 77.0% and 64.1% of the genomes consisting of repeat sequences (Supplementary Tables 4 and 5), respectively. Global methylation levels of mCG and mCHG were also higher in tetraploid genomes than in diploid and hexaploid genomes, whereas mCHH was the highest in the diploid (Supplementary Fig. 4). Chromosomal regions enriched in repeats, particularly *Gypsy* TEs, appear highly silenced, with low transcript and high mCG levels (Supplementary Fig. 5).

### Subgenome origin and polyploidization history of WBs

Subgenomes of bamboos were identified by both phylogeny-based and sequence similarity-based strategies. We assembled two syntenic gene data sets, that is 456 'perfect-copy' syntenic genes (with 1:2:3 expected copies in diploid, tetraploid and hexaploid bamboos, respectively) and 13,891 'low-copy' syntenic genes (with equal to or less than 1:2:3 copies) broadly distributed along all chromosomes (Extended Data Fig. 2a and Supplementary Fig. 6), for phylogenetic analyses. Four distinct subgenomes of WBs, that is A, B, C and D subgenomes, and H for HBs as identified previously[17], were consistently supported in analyses of both data sets (Supplementary Figs. 7 and 8; Supplementary Information). Sequence similarity analyses also supported the identification of subgenomes (Extended Data Fig. 2b,c), with subgenomes A and D clustered together.

We removed 26 outliers out of the 456 syntenic genes (Supplementary Fig. 9 and Supplementary Table 6) and recovered the monophyly of subgenome lineages of WBs (Fig. 2a and Extended Data Fig. 3a–c). Nevertheless, extensive topological discordance was present among gene trees and the coalescent-based tree and short internodes with conflicting topologies surrounded the progenitors of the A and D subgenomes, indicating the likelihood of a non-bifurcating phylogeny. Focusing on the major conflicts involving H, A and D progenitors, the most common topologies accounted for 57%, 48% and 46% of gene trees (Fig. 2b and Supplementary Table 7), respectively, which matched the bifurcating tree. Moreover, the frequencies of the other two minor alternative topologies were unequal, which was not expected under incomplete lineage sorting (ILS) alone[26], with low ILS signals (Supplementary Fig. 10). Analyses using more perfect-copy genes with subsampled species gave the same results (Supplementary Fig. 11 and Supplementary Tables 8 and 9).

We thus inferred phylogenetic networks and putative introgression events (Fig. 2c and Extended Data Fig. 3d–g) and identified hybridization between the B and C progenitors, leading to a hybrid diploid ancestor that diverged into the A and D progenitors, in accordance with the incongruent patterns of gene trees above. A second reticulation event, between the H and A progenitors, was

---

**Fig. 1 | Overview of bamboo species diversity and characterization and syntenic landscape of 11 sequenced genomes. a,** Distributions of four major clades of bamboos around the world depicted by different colored cross-hatching. The map was generated using the mapping tool ArcGIS (version 10.2; www.esri.com). **b,** Schematic representation in circles of chromosomes in 11 bamboo genomes, with rice as the outgroup. The 11 species represent the herbaceous bamboos (HBs; $2x$) and three woody clades (temperate woody bamboos, TWBs ($4x$); neotropical woody bamboos, NWBs ($4x$); and paleotropical woody bamboos, PWBs ($6x$)). Conserved syntenic blocks between circles of

H, C, B, A and D subgenomes are indicated by purple, blue, orange, light green and dark green lines, respectively. The circos diagram shows distribution of genomic features for a PWB species, *D. sinicus*. Gene expression levels in leaf tissue, mCG methylation levels, gene density and synteny between subgenomes A and B and A and C are shown from outer to inner tracts. **c,d,** Images of representative species of WBs (**c**) (scale bar = 20 m) and HBs (**d**) (scale bar = 0.3 m). The woody *D. sinicus* is the world's largest known bamboo, up to 37.5 m in height with culm diameter to 28.7 cm, and the herbaceous *Ra. guianensis* is only ~0.3 m in height.

also suggested by introgression analyses and corroborated by ~16% of the gene trees. However, introgression from other diploid ancestors of WBs to the H progenitor may have also occurred (Supplementary Fig. 12 and Supplementary Table 10), especially if these sequence signals were diluted over evolutionary time with only weak evidence remaining. Ancient hybridization between the ancestors of HBs and

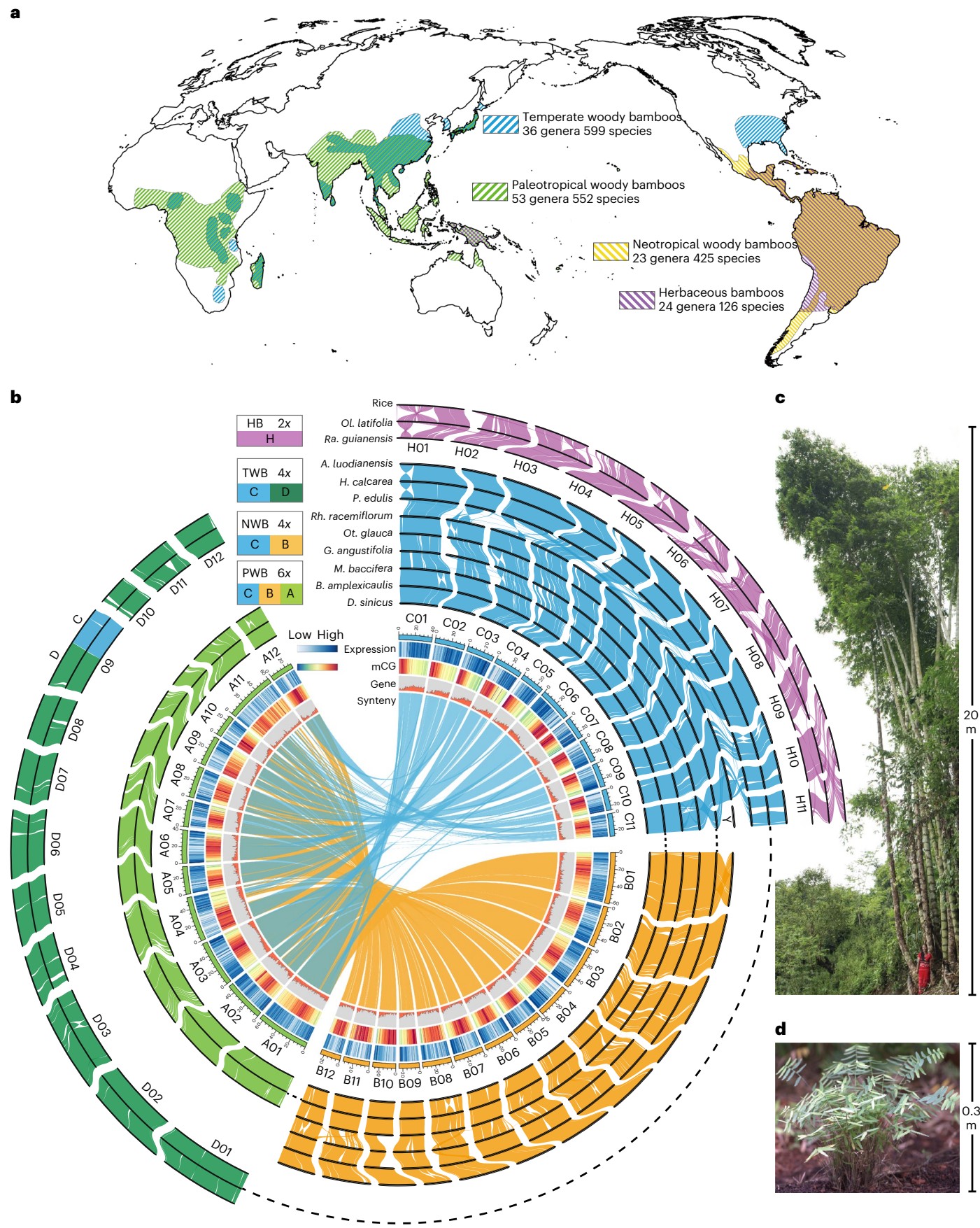

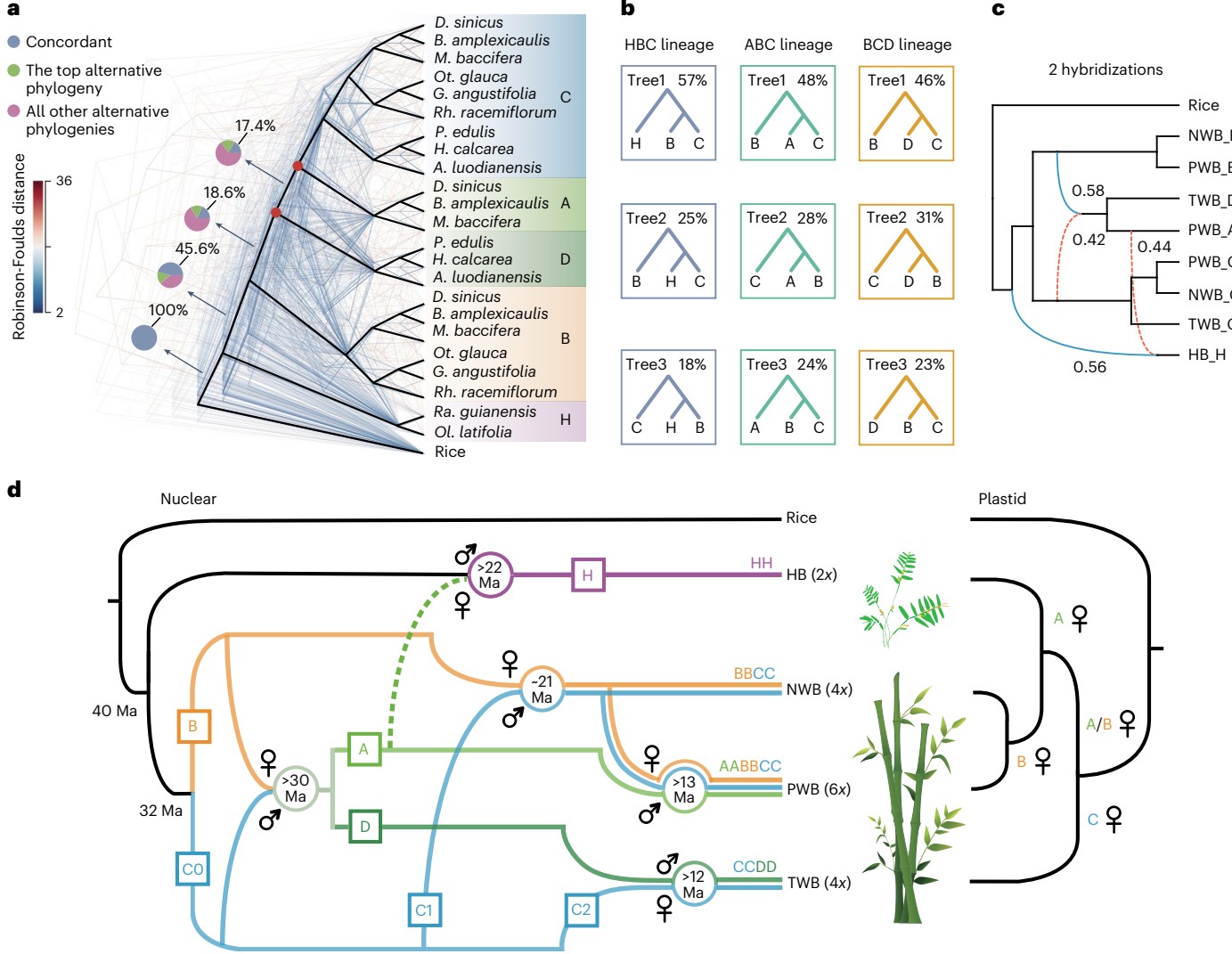

**Fig. 2 | Origin and evolution of major bamboo clades. a**, Phylogenetic analyses of bamboo subgenomes revealing massive discordance among individual gene trees. The coalescent-based tree reconstructed from 430 perfect-copy syntenic genes is shown in heavy black lines with backbone nodes without 100% bootstrap support marked in red circles. The 430 individual gene trees are colored by the Robinson-Foulds distances relative to the coalescent tree. Pie charts along the backbone phylogeny present the proportion of gene trees supporting the represented topology (blue), the main alternative topology (green) and the remaining alternatives (purple). **b**, Proportions of contrasting gene tree topologies for the 430 genes with regard to three major conflicting relationships.

**c**, Two hybridization scenarios among different diploid bamboo ancestors by PhyloNet analysis of 430 genes. Blue solid and red dashed curved lines indicate the major and minor edges that contribute to the hybrid descendants with the numbers indicating the inheritance probabilities of each parent. **d**, Model for the origins and evolutionary history of diploid bamboo ancestors and the polyploidization events in three woody clades. The five diploid progenitors (A–D and H) are indicated by different colors. Approximate dates for hybridization events are given in circles in units of million years ago (Ma). The plastid tree at the right illustrates the phylogeny of maternal donors for major bamboo clades.

WBs was also indicated by the plastid phylogeny (Supplementary Fig. 13)[27], with HBs sister to NWBs and PWBs, and by ~7% of nuclear gene trees.

Collectively, we propose a refined model for the origins and polyploidizations of bamboos (Fig. 2d). The time scales of reticulate evolution were bracketed by the divergence time of parental lineages as the upper limit and species divergence as the lower one (Supplementary Fig. 14). Differentiation of the herbaceous and woody lineages occurred early in bamboo evolution, followed by divergence of the woody ancestors into two (B and C) rather than four or five diploid progenitors[17,18]. The diploid progenitors of A and D likely originated through homoploid hybrid speciation between the B and C progenitors from 32 to 30 Ma with the former as female parent. The hybridization between the B and C1 lineages followed by polyploidization around ~21 Ma gave rise to NWBs (BBCC). With the tetraploid as maternal donor, a phenomenon also observed in

wheat and oat[28,29], the second polyploidization occurred no later than ~13 Ma, leading to the emergence of PWBs (AABBCC). The third event, also involving the C lineage (C2), led to the origin of TWBs (CCDD) before ~12 Ma.

## Karyotype stability in the evolution of WBs

Except for fission and fusion of chromosome 12 (chr12) into chr3, chr6 and chr11 in the C subgenome of NWBs and PWBs (Extended Data Fig. 2d), the four woody subgenomes have all maintained global synteny with 12 chromosomes since their divergence about 30 – 32 Ma (Supplementary Fig. 15). High-level synteny was also preserved across multiple species deriving from the shared polyploidization events (Fig. 1b), at least 12 Ma for the most recent one. However, the shortest chromosome (Y, 38.9 Mb) in *Rh. racemiflorum* has no homoeolog, as well as lower gene density and expression than other chromosomes (Supplementary Fig. 16); it could be a B chromosome[30], requiring further

investigation. Reconstruction of ancestral bamboo karyotypes (ABKs) also revealed that woody subgenomes, particularly A, B and D, resembled the ancestral grass karyotype (AGK)[22], maintaining stunning evolutionary stability over a long period of evolution (Fig. 3a). Large-scale rearrangement among subgenomes was only found for a mosaic chromosome formed by fusion of chr9D and a large segment (38.9–54.8 Mb) of chr2C (Extended Data Fig. 2d), which was shared by three TWB species, indicating the occurrence prior to species divergence. Putative homoeologous exchange was also found at a low level of 0.43% to 1.27% of genes for subgenomes in WBs (Supplementary Fig. 17 and Supplementary Table 11). By contrast, many rearrangements were found in HBs (Fig. 3a), including a chr10–chr12 fusion and accompanied chromosome number reduction.

Most fission and fusion events occurred in the H and C subgenomes (Fig. 3b and Supplementary Table 12). However, these events in HBs were largely species-specific, with only three of 36 ones shared by two species. By contrast, many in the C subgenome were shared by different species within the tropical and temperate clades, respectively, suggesting a possible role of polyploidization in inducing genomic rearrangements despite general karyotype stability. Additionally, different patterns were observed between tropical and temperate clades (Fig. 3b), consistent with the divergence of C into C1 and C2 in independent polyploidizations. However, the addition of the A subgenome had little impact on the rate or nature of subsequent rearrangements in PWBs.

We identified 1,494 inversions (>1 kb) in 11 bamboo genomes (Supplementary Table 13). Once more, HBs tend to contain a larger number of species-specific inversions. Within WBs, the C subgenome experienced the fewest but also large inversions (>10 Mb) with the longest total length (Extended Data Fig. 4a). We traced the evolution of shared inversions (Supplementary Table 14) and found that most occurred at nodes after polyploidization prior to species divergence (Fig. 3a). Notably, eight inversions were shared only by the A and D subgenomes, confirming their origin from a common ancestor.

### Divergent trajectories of subgenomes

As demonstrated above, the C subgenome stood out among the four subgenomes of WBs. It was also smaller than the A and B subgenomes but similar to the D subgenome in size, closely correlated with the TE content (Extended Data Fig. 2e,f). The larger subgenomes (average 784.2 Mb in TWBs and 721.1 Mb in NWBs versus 345.3 Mb in PWBs) made the tetraploid genomes substantially larger than those of the hexaploids. The smaller size of the hexaploid genomes was mainly due to the lower percentage of *Gypsy* elements (14.1% versus 28.0% in tetraploids). These results indicate varied TE dynamics among subgenomes as well as tetraploid and hexaploid clades following polyploidizations.

Gene evolution can be abruptly altered by polyploidization, with many whole-genome duplicates subject to extensive loss[31], as found in WBs here (Fig. 3c and Supplementary Fig. 18). Moreover, a gene retention level of C > B/D was observed in tetraploids, while a pattern of A > B > C was recovered in PWBs, suggesting variable patterns of biased fractionation among subgenomes in tetraploids and hexaploids. The fractionation pattern was also validated by excluding the possibility

of mis-assemblies of single- and multicopy genes (Supplementary Fig. 19). With genomes of five representative grasses and 11 bamboos (Methods), we found that 50.0% to 77.5% of the genes of the subgenomes in WBs were present in homoeologous groups (Extended Data Fig. 4b and Supplementary Table 15). Most groups (74.1%–85.1%) were maintained as 1:1 in tetraploids; many fewer were retained as 1:1:1 in hexaploids (21.8%–25.2%). The C subgenome had more conserved subgenome-specific genes and thus more genes in total within the tetraploid genomes (Supplementary Table 16); however, it was the A subgenome having the most genes in hexaploids. The number of core grass gene families present in all 16 analyzed genomes was greater in the A and C subgenomes in hexaploid and tetraploid genomes (Extended Data Fig. 4d,e), respectively. However, gene density was consistently higher in the C subgenome (Extended Data Fig. 4c) with lower levels of TE density and methylation around genes compared to the other subgenomes in WBs (Fig. 3d and Supplementary Fig. 20). These results together imply that the C subgenome is dominant in two tetraploid clades, whereas inclusion of the A subgenome altered this dominance in hexaploid bamboos.

### Subgenome dominance and shift in WBs

To capture alterations of the transcriptional landscape after polyploidization, we sequenced and analyzed 476 transcriptome samples representing different tissues at various developmental stages across the 11 sequenced bamboos (Supplementary Table 17), mostly with three biological replications per tissue per species (Supplementary Fig. 21). In WBs, genes have lower expression breadth across tissues, compared to those in HBs (Supplementary Table 18), pointing to subgenome expression divergence. Compared to the other three subgenomes in WBs, the C subgenome always has a higher proportion of expressed genes (Supplementary Table 19), as well as the highest average expression level (Extended Data Fig. 4f).

To determine expression patterns of subgenomes in each clade, we identified 4,123 and 3,839 1:1 homoeologous gene pairs across subgenomes shared by all three TWB and NWB species, respectively, and 1,157 triads (1:1:1) for PWBs. Principal-component analysis (PCA) showed clear separation of expression between tissues (PC1 and PC2), followed by clear separation by subgenomes (PC2 and PC3) in all three clades (Extended Data Fig. 4g). This separation was also observed in analyses of individual species with more homoeologous genes (Supplementary Fig. 22). Subgenomes showed consistent patterns of up- or down-regulation of genes among homoeologs across tissues and species in the two tetraploid clades while varying widely, resembling a mosaic, in PWBs (Fig. 4a and Supplementary Fig. 23). Homoeologs were further clustered into 10 groups based on their expression patterns (Supplementary Fig. 24). More than half of gene pairs (58.5%–63.5% in TWBs and 66.9%–68.1% in NWBs) and a majority of triads (82.7%–88.9%) diverged into distinct groups (Fig. 4b and Supplementary Table 20).

Comparison of expression patterns in *P. edulis* and *G. angustifolia*, as representatives of TWBs and NWBs, respectively, showed that the C subgenome had more up-regulated genes than the D or B subgenomes (*P* < 0.05, Wilcoxon rank-sum test) (Fig. 4c and Supplementary

**Fig. 3 | Structural characteristics and evolution of bamboo genomes.**
**a**, The reconstructed ABK at key evolutionary nodes. Occurrences of inversions (>1 kb) are mapped on a network with the circle size on the nodes and the terminal branch thickness proportional to the number of shared and species-specific inversions, respectively. An example of an inversion shared by the C subgenome is shown in the dotplots beneath. ABK-W, -H, -B, -C, -X refer to ABKs at different evolutionary stages, -W, -H, -B, -C for progenitors of WBs, herbaceous bamboos, subgenomes B and C, and -X for the progenitor of subgenomes A and D, respectively. **b**, Large-scale chromosomal rearrangements across 11 bamboo genomes in comparison to the rice genome and the numbers per chromosome indicated in the heat map. The numbers on nodes and branches indicate those of shared and species-specific fission (red) and fusion (blue) events, respectively.

**c**, Gene retention patterns among bamboo subgenomes relative to the rice genome. The significance of differences for interspecies comparisons (red) and intersubgenome comparisons (black) was determined by two-sided Wilcoxon rank-sum test (boxplots: centerline, median; box limits, first and third quartiles; whisker, 1.5x interquartile range; *n* is the number of sliding windows used in evaluating gene retention). **d**, Comparison of TE density and methylation levels surrounding genes among subgenomes in *P. edulis*, *G. angustifolia* and *D. sinicus*, all with significant (*P* < 0.001) differences (two-sided Wilcoxon singed-rank test) except for mCHH of upstream region of genes in *P. edulis*, mCHH of gene body in *G. angustifolia* and mCHG of gene body between the A and B subgenomes in *D. sinicus*. TSS, transcription start site; TTS, transcription termination site.

Table 21). Furthermore, this bias is consistent across all tetraploid bamboos for nearly all sampled tissues and it is more likely to occur in NWBs compared to TWBs (Extended Data Fig. 5a,b and Supplementary Figs. 25 and 26). Investigating bias is not as straightforward in the hexaploid genome[32] and we initially calculated relative transcript abundance of subgenomes. We found that the C subgenome (34.7%)

accounts for more than the A (32.8%) and B (32.5%) subgenomes in the early-diverging *M. baccifera* ($P < 0.01$, Wilcoxon rank-sum test) but not in the other two PWB species (Extended Data Fig. 5c and Supplementary Table 22), indicating a possible dominance of the C subgenome in early (but not later) PWB evolution. Moreover, the numbers of up-regulated genes are similar between the A and C subgenomes in *B. amplexicaulis*

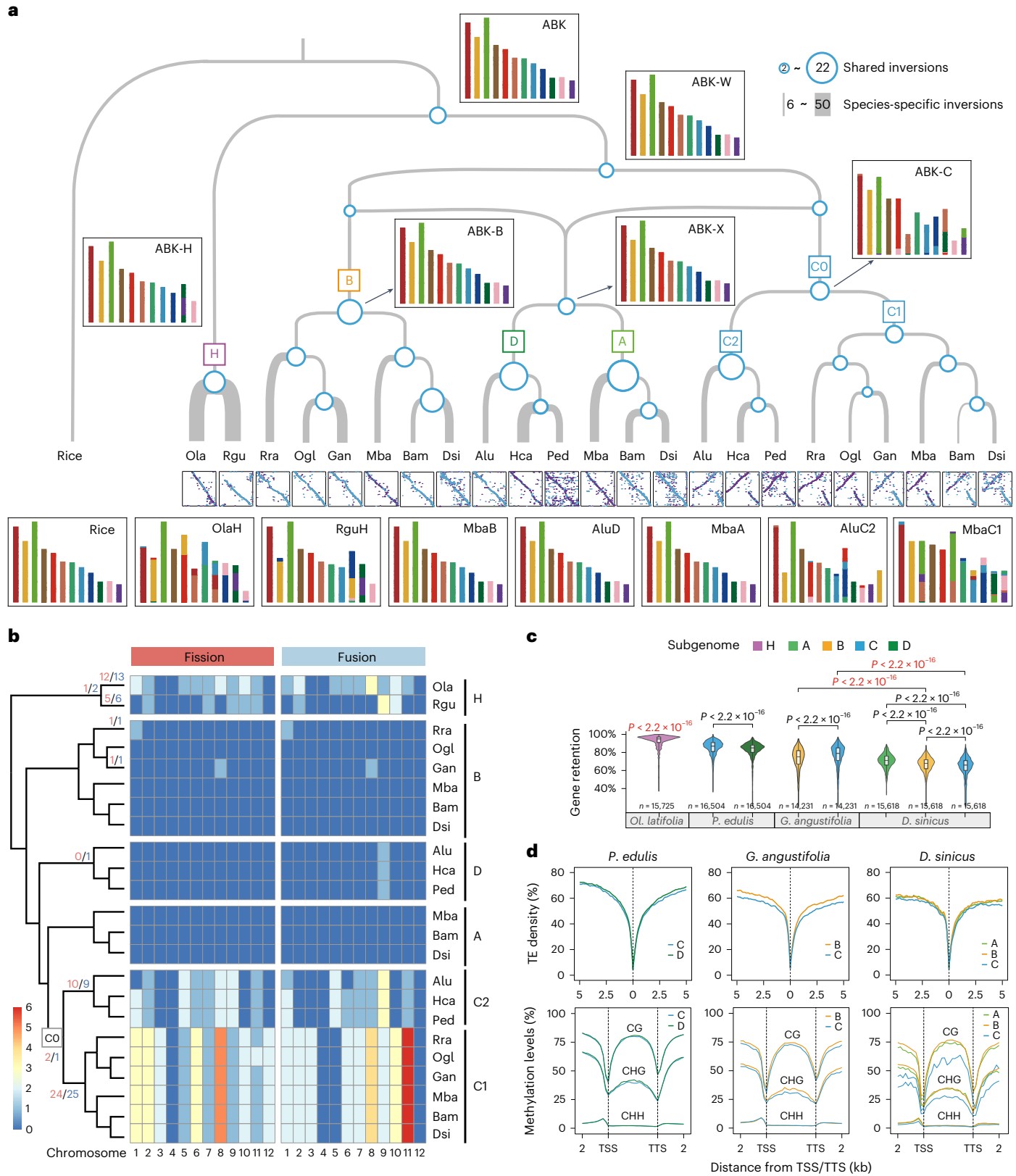

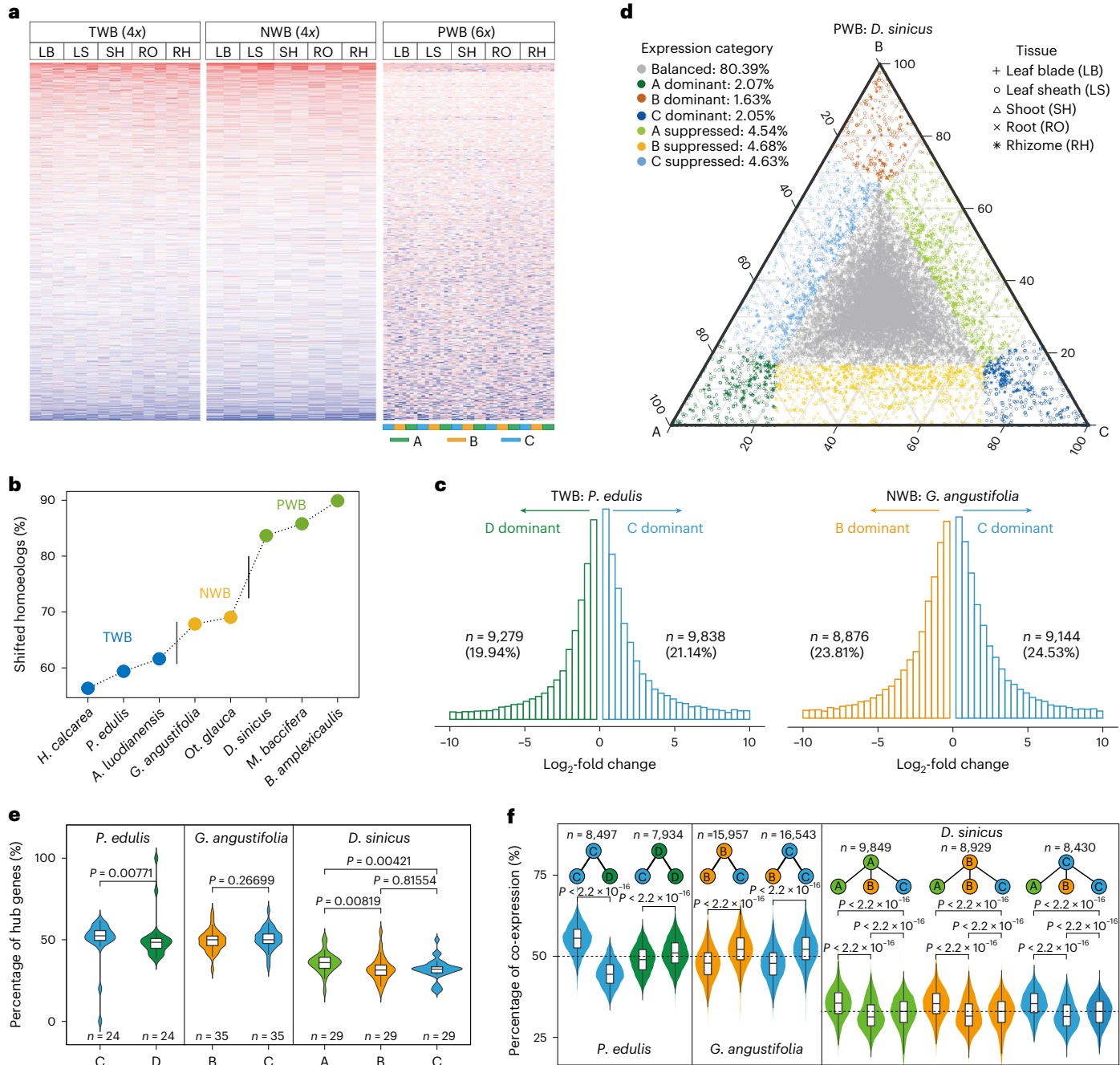

**Fig. 4 | Homoeolog expression patterns in polyploid bamboos. a**, Heat map of expression of homoeologs across five common tissues sampled in three clades of WBs: leaf blade (LB), leaf sheath (LS), shoot (SH), root (RO) and rhizome (RH). For the tetraploids, the expression of C > D or C > B genes and B > C or D > C genes is shown in red and blue, respectively. For the hexaploids, greater expression is indicated in red and lesser in blue. Each row represents one homoeologous gene pair and each column a species; a full-version is in Supplementary Fig. 22. **b**, Divergence of expression of homoeologs in three clades of WBs. Homoeologs clustered into 10 groups based on their expression patterns and those into different groups defined as shifted in expression. **c,d**, Histograms (**c**) and ternary plot (**d**) of homoeologs for expression bias in representative species

of three clades of WBs. Histograms indicate the total number of up-regulated genes across five common tissues in tetraploids. Each point in the ternary plot represents a gene triad with an A, B and C coordinates. Triads in vertices indicate dominant categories, whereas triads near edges and between vertices are suppressed categories; balanced triads shown in grey. **e**, Distribution of hub genes in the WGCNA modules for subgenomes (*n* is the number of modules). **f**, Comparison of percentage of co-expression between intra- and intersubgenome genes in the WGCNA modules (*n* is the number of genes from one certain subgenome). In panels e and f, *P* values were determined by two-sided Wilcoxon rank-sum test (boxplots: centerline, median; box limits, first and third quartiles; whisker, 1.5x interquartile range).

and *D. sinicus* ($P > 0.05$, Wilcoxon rank-sum test) (Supplementary Fig. 27 and Supplementary Table 23), despite varying biases across tissues (Extended Data Fig. 5d and Supplementary Fig. 28). However, both the A and C subgenomes have more up-regulated genes than the B subgenome in all three PWB species ($P < 0.05$ for all comparisons except for C versus B in *D. sinicus*, Wilcoxon rank-sum test).

We further considered six homoeologous expression categories[32] in PWBs (Fig. 4d and Supplementary Figs. 29 and 30). The balanced expressed triads were most common in all of the tissues of the three species (59.2%–94.9%), except leaf sheath (Extended Data Fig. 5e, Supplementary Fig. 31 and Supplementary Table 24). Triads with single-homoeolog dominance were infrequent (5.5%, 8.5% and 6.1%

in *M. baccifera*, *B. amplexicaulis* and *D. sinicus*, respectively), whereas those classified as single-homoeolog suppressed were more common (17.1%, 20.8% and 15.9%). Across tissues, the B-dominant category (1.7%, 2.6% and 1.9% in *M. baccifera*, *B. amplexicaulis* and *D. sinicus*, respectively) is lower than the A- (2.0%, 3.0% and 2.1%) or C-dominant (1.8%, 2.9% and 2.1%) category, whereas the B-suppressed category is generally larger (6.1%, 6.9% and 5.5% versus 5.6%, 6.8% and 5.1% (A) or 5.4%, 7.1% and 5.3% (C)) (Extended Data Fig. 5f). No significant difference in biased categories existed between the A and C subgenomes, and only the A-suppressed category is slightly less than the C-suppressed category in *D. sinicus* ($P = 0.04785$, Wilcoxon rank-sum test), pointing to a bias toward A relative to the C subgenome in it.

To determine whether genes of the biased subgenome are more likely to be co-expressed, we performed weighted gene co-expression network analyses (WGCNA)[33] for *P. edulis*, *G. angustifolia* and *D. sinicus* as representatives of WBs with broad transcriptomic sampling, and *Ra. guianensis* for HBs, with 24 to 50 modules identified (Supplementary Table 25). More genes were co-expressed from the C compared to B and D subgenomes in tetraploids (Extended Data Fig. 6a). More importantly, hub genes in the networks were also overrepresented in the C subgenome (Fig. 4e and Extended Data Fig. 6b). In contrast, in the hexaploid *D. sinicus*, the A subgenome instead had more hub genes. Furthermore, genes are more likely to be co-expressed with C-subgenome genes in *G. angustifolia*, whereas co-expression was more frequently found among genes from the same subgenome in *P. edulis* (Fig. 4f). In *D. sinicus*, co-expression with A-subgenome genes was the most frequent, both within and between subgenomes, followed by co-expression with the C and then B subgenomes. These results further support the dominance of the C subgenome in both the TWB and NWB clades with independent origins, whereas dominance appears to have shifted gradually from C to the A subgenome during PWB evolution. Moreover, dominant expression could have formed shortly following the polyploidizations and continuously accumulated in WBs (Extended Data Fig. 6c and Supplementary Table 26).

## Genomic variation and the origin of unique traits in WBs

Within Poaceae, WBs have evolved unique traits that include lignified culms and infrequent flowering (Fig. 5a). The shoot was the most distinctive tissue in WBs but not in HBs, based on gene expression (Extended Data Fig. 7a, b and Supplementary Fig. 32), suggesting an evolutionary innovation of shoot in the rapidly growing WBs. Moreover, expression similarity clustered the root and rhizome together and also the shoot and culm leaf sheath (homologous to foliage leaf sheath) together.

To uncover the genomic basis of the origin of exceptional traits in WBs, we investigated gene family size, new genes and positively selected genes (PSGs) during their evolution (Fig. 5a). We also identified shoot- and inflorescence-specific expressed genes (Supplementary Table 27) with 1,349 genes shared by *P. edulis* and *D. sinicus*. In all, 163 new gene families accompanied the origin of WBs (Supplementary Table 28). Of these, 32 and 19 were specifically expressed in the shoot of *P. edulis* and *D. sinicus*, respectively, with a generally higher transcriptome age index (TAI) for the C subgenome (Extended Data Fig. 7c, d and Supplementary Fig. 33a), suggesting functional roles of new genes[34], particularly those of the C subgenome, in the shoot. A total of 6,800 gene families were significantly expanded with the polyploid origins of WBs (Supplementary Fig. 34 and Supplementary Table 29), although tandem and dispersed duplications also played a role (Supplementary Table 30). Genome-wide screening revealed 183 PSGs shared by all three polyploid clades (Supplementary Fig. 35, Supplementary Tables 31 and 32), with those from the C subgenome enriched. Moreover, the genes experiencing two or more genomic changes above had overrepresentation of the C subgenome (Fig. 5a and Supplementary Fig. 33b). Many of them potentially involved in the unique life cycle of WBs, such as *GI* and *SPL7* as key regulators of flowering[35], were all from the C subgenome.

Functional enrichment analyses showed that expanded gene families, at the whole-genome and subgenome levels, particularly for the C subgenome, were mainly associated with plant vegetative growth and development (for example, 'plant hormone signal transduction' and 'phenylpropanoid biosynthesis') (Fig. 5b). Another notable term, 'circadian rhythm', is enriched in flowering signal genes. Intriguingly, shared PSGs were also enriched in similar functional terms.

We further investigated genomic changes in the lignin biosynthesis pathway[36] (Fig. 5c) for insights into their contributions to bamboo woodiness. Shoot growth of *D. sinicus*, which can reach 10 m of height in 30 days, shows a 'slow-fast-slow' pattern as in other WB species[14,37], with four stages defined (Extended Data Fig. 8a–c). Lignin, cellulose and hemicellulose were deposited synchronously (Supplementary Table 33), ensuring mechanical support for the fast-growing shoot. Nearly all lignin-related genes have expanded copies through polyploidy-derived duplicates[38] in WBs compared to HBs and grasses (Supplementary Table 34), and tandem duplication was further observed as for *COMT* and *F5H1* in *D. sinicus*. Thirty-one genes in the pathway with a majority experiencing some kind of genomic changes (Fig. 5c) were detected as positive regulators of shoot growth in *D. sinicus* (Extended Data Fig. 8d,e). The most notable was *COMT*, playing a key role in the lignification of the giant *D. sinicus* shoot (Extended Data Figs. 8f and 9a,b) and being mainly responsible for biosynthesis of S monolignol[39], which is critical for the strength of culm in the grasses.

Except for loss from the B subgenome in two species, all bamboo *COMT* copies occur in a conserved syntenic region corresponding to rice chr8 (Fig. 5d and Extended Data Fig. 9d). However, the segment containing *COMT* (comprising ~165 genes in tetraploids and ~116 genes in hexaploids) was translocated from chr8 to chr9 in the C subgenome, indicating an event possibly underlying the adaptive evolution of this gene by positive selection in the common ancestor of WBs (Extended Data Fig. 9c). Additionally, its expression in the shoot was generally dominated by the C copy in tetraploid bamboos and *M. baccifera* (Fig. 5d). In the two remaining PWB species, the A copies accounted for more than two thirds of the total expression, consistent with the general trend of dominance shifting from C to the A subgenome in PWB evolution. Positive selection and biased expression of *COMT*-C may represent a first step in the evolution of bamboo woodiness, and subsequently, the shift of biased expression and tandem duplication of *COMT*-A was probably associated with *D. sinicus* evolving into the world's largest known bamboo.

We found larger *Ka*/*Ks* (nonsynonymous to synonymous nucleotide substitution) values in WBs compared to HBs (Extended Data Fig. 9e), indicating an overall relaxed selection of genes in WBs. Moreover, selection on genes exclusively expressed during reproduction was relaxed further than selection on genes confined to the vegetative stage in WBs (Extended Data Fig. 9f), whereas no difference was found in HBs. Overall, these genomic changes that accompanied polyploidization and dynamic subgenome dominance highlight the genomic basis of the evolution of unique traits and associated adaptation of WBs.

## Discussion

Using multiple genome assemblies for each clade, we resolved the reticulate evolution of bamboos[16–18] by identifying and tracing four ancient subgenomes of WBs (that is, A, B, C and D) and the genome of HBs (H). Recurrent hybridization events between diploid ancestors of woody lineages followed by polyploidization, together with introgression between ancestral woody and herbaceous lineages, occurred deep in the evolution of bamboos. Our results demonstrate not only how hybridization and polyploidization generated deep conflicting phylogenies but also their roles as driving forces in species diversification[2,3,40], as seen in the contrasting numbers of documented species in WBs (1,576) versus HBs (126). With two independent tetraploidization events and hexaploidization involved in the origin of major clades, the WBs represent a remarkable polyploid system exhibiting karyotypic

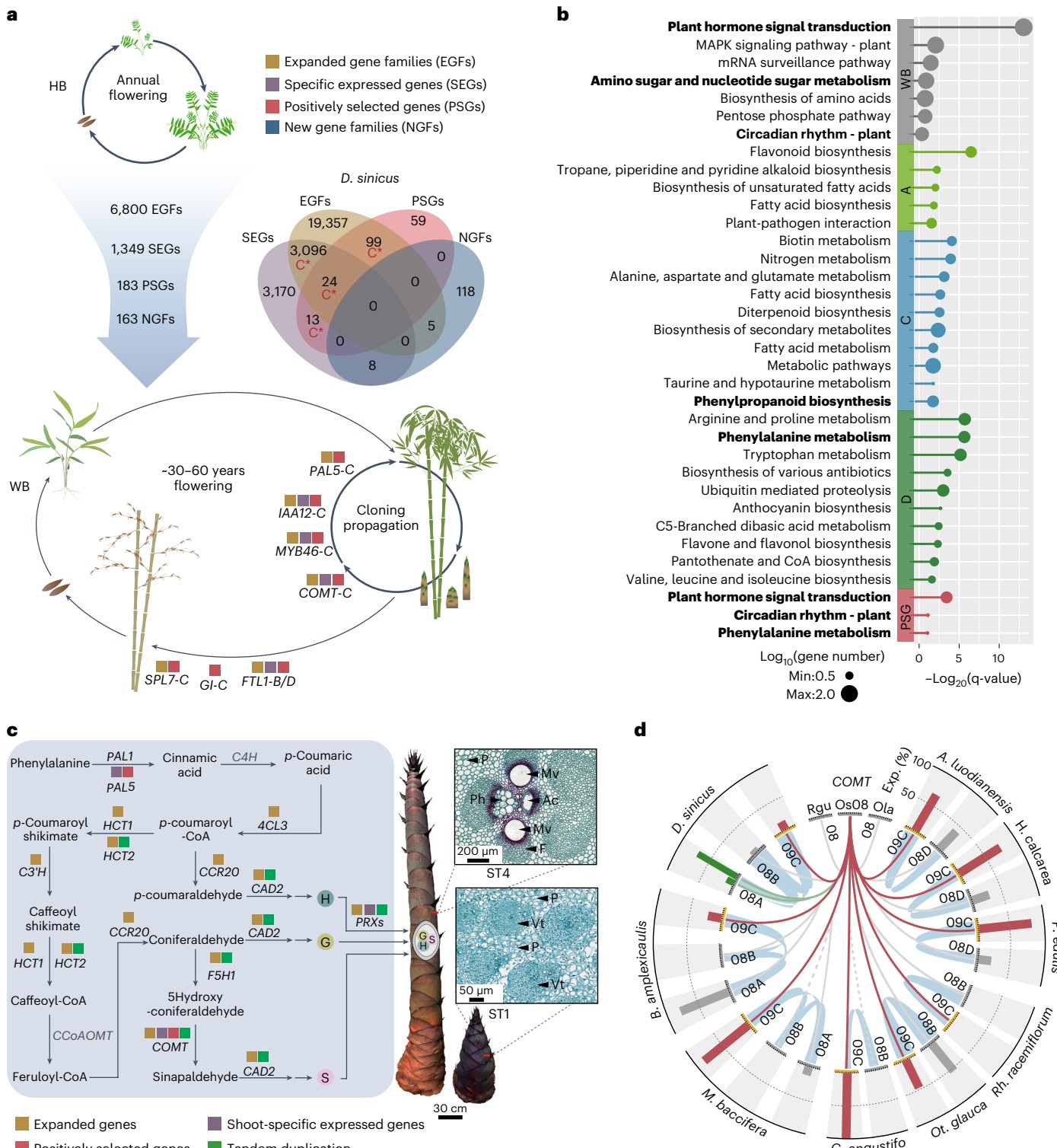

**Fig. 5 | Genomic basis for the evolution of WBs. a**, Diagram of the life history transition from herbaceous (HB) to woody (WB) bamboos with underlying genetic alterations. Venn diagram shows identified genes from expanded gene families (EGFs), specific expressed genes (SEGs), positively selected genes (PSGs) and new gene families (NGFs) in *D. sinicus* with those overrepresented in the C subgenome indicated by a star. **b**, The KEGG enrichment of expanded gene families in WBs and three subgenomes (no significant result for the A subgenome) individually, as well as of PSGs shared by WBs, with enriched terms closely associated with the unique traits of WBs indicated in bold. **c**, Genetic alterations in the lignin biosynthesis pathway and morphological/anatomical observations of *D. sinicus* shoot during fast growth (stage 1 (ST1) and stage 4 (ST4); the whole four stages in Extended Data Fig. 8b,c). Genes in black are identified as positively

correlated with lignin biosynthesis in shoot growth of *D. sinicus*, while those not positively correlated are shown in gray. H, G and S represent the H units, G units and S units of lignin, respectively. Scale bars = 30 cm (shoot), 50 μm (micrograph at ST1) and 200 μm (micrograph at ST4). P, parenchyma cells; F, fiber cells; Ph, phloem; Mv, metaxylem vessel; Ac, cavity formed by the degradation of protoxylem; Vt, vascular tissue. **d**, Syntenic relationships and expression of *COMT* in the fast-growing shoot of WBs. Synteny between *OsCOMT* and its homologous bamboo genes is connected by curves with those in dark red indicating transition from chr8 to chr9 for the C-subgenome copy; gene losses indicated in dashed lines. Bar in the outer circle shows the relative contribution percentage of each subgenome to the overall expression of *COMT* in individual genomes. Os, *Oryza sativa*; Ola, *Olyra latifolia*; Rgu, *Raddia guianensis*.

stasis without cytological dysploidy, despite 12 to 20 Ma since polyploidization and subsequent large-scale species diversification. Bamboos thus provide a rare opportunity to study the long-term effects of polyploidization and the evolution of subgenome dominance, in contrast to recent polyploids without large-scale species diversification[6,9–11] or ancient polyploids that have already experienced massive subgenome reshuffling[41].

Although the prevalence of subgenome dominance is a matter of discussion[7–9,11], our analyses suggest unambiguously dominant subgenomes in polyploid bamboos, as reflected in a series of features including genomic rearrangements, gene fractionation and gene expression, among others. However, the pattern of dominance at the expression level is more dynamic, particularly in the hexaploid bamboos. Furthermore, subgenome dominance could be established shortly after polyploidization[42], as is the case in NWBs and TWBs, and inherited by their descendants. The parallel origin of C subgenome dominance in the two tetraploid clades was likely to be related to its genome architecture (for example, TE density and methylation patterns), as in other polyploid genomes[4,42]. Intriguingly, dominance can be shifted with the integration of a new subgenome as shown in the hexaploid clade. The dominant C subgenome, together with the A subgenome in the hexaploid clade, contributed the most to the evolution of distinctive traits in WBs and possibly their adaptive radiation into forest habitats. In turn, the life history transition from annual flowering in HBs to long flowering cycles in WBs and thus less chance of rearrangement during meiosis might be one of the reasons explaining the observed minimal subgenome reshuffling. This transition, coupled with polyploidization, has also likely reshaped the evolution of subgenomes with relaxed selection. Finally, our work highlights the utility of using clade-wide genome assemblies to advance our understanding of subgenome evolution in polyploids. Further efforts on similar evolutionary scales are needed to test the generality of the present findings across the green plant kingdom.

## Online content

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

## Methods

### Plant materials, sequencing and assembly

Eleven bamboo species representing all four major clades of Bambusoideae were selected for genome sequencing and large-scale transcriptome sequencing. Briefly, genomic DNA from 11 bamboo species was firstly used for short-read sequencing (150 bp). Genome size and heterozygosity were estimated using a *k*-mer-based approach by GenomeScope[43] with default settings. Subsequently, for the 11 genomes, high-quality genomic DNA was sequenced by the Oxford Nanopore Technology (ONT). Hi-C libraries were constructed following a published protocol[44] and sequenced.

The ONT long reads were self-corrected using CANU (v1.7)[45] with default values and further assembled into contigs using SMARTdenovo v1.0.0 (https://github.com/ruanjue/smartdenovo) with default parameters or NextDenovo v2.3.1 (https://github.com/Nextomics/NextDenovo) with 'reads_cutoff: 1k and seed_cutoff: 31k'. Then, corrected ONT long reads were used for three rounds of initial polishing by Racon (v1.4.21)[46] or Nextpolish (v1.3.0)[47] with default parameters, and short reads were further applied for three rounds of correction using Pilon (v1.23)[48] or Nextpolish (v1.3.0)[47].

The Hi-C sequencing data were mapped to polished contigs using BWA (v0.7.10-r789)[49] with '-aln' or Bowtie2 (v2.3.2)[50] with '-end-to-end,-very-sensitive -L 30', and only uniquely mapped read pairs with mapping quality of more than 20 and valid interaction read pairs filtered by the HiC-Pro (v2.8.1)[51] were retained for further analysis. The polished contigs were then scaffolded, ordered and anchored into pseudo-chromosomes using filtered Hi-C data by LACHESIS software[52].

### Assembly quality evaluation

The contiguity and completeness of the genome assemblies were assessed by two approaches. First, short paired-end reads were mapped to their corresponding genomes using BWA (v0.7.10-r789)[49] with default parameters. Second, assembly contiguity was assessed by LTR Assembly Index (LAI)[23] following the standard of Draft: 0 ≤ LAI < 10, Reference: 10 ≤ LAI < 20, and Gold: 20 ≤ LAI. We further used calculate_AG in Mabs (v2.19)[25] (–local_busco_dataset Poales_odb10) to determine the count of accurately assembled genes (AG). The AG values are calculated by summing the number of genes in both single- and true multicopy BUSCO orthogroups by distinguishing true from false ones based on sequencing coverage.

### Annotation of genomes

The repeat sequences of the 11 bamboo assemblies were identified by Extensive de novo TE Annotator (EDTA) (v1.8.5)[53]. LTR retrotransposons were predicted using LTR_Finder (v1.07)[54] and LTR_retriever (v2.6)[55]. TIR transposons were identified using an integrated strategy with Generic Repeat Finder (v1.0)[56] and TIR-Learner (v1.19)[57], and Helitron transposons were identified by HelitronScanner (v1.1)[58]. All the programs were performed with default parameters. LINEs were detected by RepeatModeler v2.0.1 (https://github.com/Dfam-consortium/RepeatModeler). The curated TE library (rice 6.9.5.liban) of EDTA was used to annotate repeat sequences with parameters '–species others–step all–sensitive 1–evaluate 1–anno 1'.

Protein-coding gene models were predicted by integrating three strategies: ab initio prediction, homology-based search and expression evidence. The ab initio prediction was conducted using Genscan[59], Augustus (v2.4)[60], GlimmerHMM (v3.0.4)[61], GeneID (v1.4)[62] and SNAP (v2006.07.28)[63] with default parameters. The GeMoMa (v1.3.1)[64] was applied for homology-based gene annotation using genomes of *Arabidopsis thaliana* (https://www.arabidopsis.org), rice (MSU V7.0) and sorghum (*Sorghum bicolor*) (Gramene V60). RNA sequencing (RNA-seq) reads obtained from leaf of each species were aligned to the corresponding assemblies using HISAT2 (v2.0.4)[65] with parameter '-max-intronlen 20000, -min-intronlen 20' and Stringtie (v1.2.3)[66] to generate predicted transcripts. The resulting transcripts were passed to

TransDecoder v2.0 (https://github.com/TransDecoder/TransDecoder) and GeneMarkS-T (v5.1)[67] for prediction of protein-coding regions. Finally, the consensus gene models were generated by EvidenceModeler (v1.1.1)[68] and refined using PASA (v2.0.2)[69]. The BUSCO v4.0.6 pipeline[70] was used to estimate the completeness in genic regions using the Poales_odb10 database.

### Bisulfite sequencing and methylation analysis

We selected four bamboo species (*Ra. guianensis*, *P. edulis*, *G. angustifolia* and *D. sinicus*) representing HBs, TWBs, NWBs and PWBs, respectively, for whole-genome bisulfite sequencing. Two biological replicates were collected for each leaf sample. Whole-genome bisulfite sequencing libraries were sequenced with paired-end reads of 150 bp and clean reads were mapped to the reference genome using Bismark (v0.21.0)[71] with default parameters. The bisulfite conversion rate above 99.8% in all samples was estimated by lambda genome methylation levels. The genome-wide methylation level was obtained using ViewBS (v0.1.9)[72]. For gene methylation analyses, the gene body and 2-kb regions upstream and downstream were divided into 50 and 40 bins, respectively.

### Subgenome identification

Phylogenetic tree-based and sequence similarity-based strategies were adopted for subgenome identification. For the tree-based approach, two genome-wide syntenic gene data sets; that is, perfect-copy and low-copy syntenic genes were extracted from syntenic blocks across 11 bamboo genomes and the rice genome. The syntenic blocks were generated by the jcvi (v1.1.17)[73] with the '–quota' parameter set to 1, 2 and 3 for the diploid, tetraploid and hexaploid bamboo genomes. In total, 456 perfect-copy syntenic genes from 29 blocks and 13,891 low-copy syntenic gene clusters from 41 blocks were obtained.

The coding sequences of genes were aligned using MAFFT (v7.471)[74] and then converted into amino acid sequences and trimmed using PAL2NAL (v14)[75] under '-nogap -nomismatch'. Concatenation matrices of perfect-copy gene alignments were generated for each syntenic block. Maximum likelihood (ML) trees for each concatenation and individual gene alignment were inferred using RAxML (v8.2.12)[76] under the GTRGAMMA model with 200 rapid bootstrap replicates. Protein sequences of low-copy syntenic genes for each block were passed to OrthoFinder (v2.3.12)[77] to infer orthogroups and generate the phylogeny of species.

For the sequence similarity-based strategy, pairwise comparisons were made between different subgenomes of WBs and genomes of HBs. 1:1 syntenic gene pairs between all comparisons were generated, and global similarity of each pair was calculated using Identity (v1.0)[78] with a threshold >0.6.

### Phylogenetic analysis

To decipher the phylogenetic relationships among subgenomes, we identified outlier genes and filtered the 456-gene data set (Supplementary Information). 430 remained perfect-copy syntenic genes were concatenated and fourfold degenerate sites were extracted using MEGA-X[79] for inference of ML trees as described above and the coalescent-based tree by ASTRAL (v5.6.3)[80] (-i <gene trees > -t 3). Divergence times among subgenome lineages were also estimated with the concatenated 430-gene data set.

We built the ML tree based on the 11 bamboo plastomes and also assembled a larger data set of 2,021 perfect-copy syntenic genes for analyses (Supplementary Information). Gene tree discordance within the 430 and 2,021 genes was quantified and visualized by drawing cloud trees for all gene trees using the ipyrad analysis toolkit (v0.9.74)[81]. Nodes with <50% bootstrap support were collapsed by Newick utilities (v1.6.0)[82], and then phyparts (v0.0.1)[83] (-a 1 -v -o) was used to summarize the conflict and concordance information between the gene trees and the coalescent tree.

## ILS, hybridization and introgression analyses

To detect the underlying causes of incongruent phylogenetic patterns, the theta parameter reflecting the level of ILS[84] for each internal branch of the 430-gene data set was evaluated by dividing the mutation units inferred from RAxML and coalescent units inferred from ASTRAL. Network analyses were carried out using PhyloNet (v3.8.0)[85] for both the 430- and 2,021-gene data sets with the Infer_Network_MPL method under '-o -pl 20 -b 50 -x 50'. For the 430-gene data set, the same subgenomes across different species were associated using an additional '-a' parameter to reduce the computational burden. Three parallel network searches with zero to two reticulation events were performed. To infer putative introgression events, we ran QuIBL[86] for each triplet under default values with the 430 gene trees as input. Additionally, we conducted HyDe (v0.4.3) analysis[87] using the concatenated alignment of the 430-gene data set, and the same subgenomes from different species were regarded as different replicates.

## Ancestral karyotype reconstruction

Four species were chosen to trace the evolution of the bamboo karyotype—*Ol. latifolia*, *Ra. guianensis* and two early-diverging woody species (*A. luodianensis* and *M. baccifera*), which together contain all of the subgenome types. First, the HB genomes and woody subgenomes, with the rice genome as reference, were aligned to each other using MCScan software[88] with the '–quota' parameter set to 1, and 1:1 syntenic homologs were identified. Second, conserved syntenic blocks were filtered and extracted using DRIMM-Synteny[89] with default values. Third, ancestral genome structure at key evolutionary nodes were reconstructed using the IAGS program[90] with the GMP model.

## Identification of genomic rearrangements and putative HEs

Based on the chromosome-level synteny generated above, the fusion and fission events in the 11 bamboo genomes compared with the rice genome were determined. Alignments between rice and bamboo chromosomes were generated using the nucmer program embedded in MUMmer (v4.00rcl)[91] with default parameters, then passed to the delta-filter program to retain highly reliable alignments with length ≥100 bp and identity ≥80%. Breakpoints for fusions and fissions were identified based on the resulting syntenic coordinates, and common events shared by subgenomes were identified by comparing two breakpoints using bedtools (v2.30.0)[92].

To detect inversions (>1 kb) in the 11 bamboo genomes, all bamboo chromosomes were oriented using EMBOSS (v6.6.0)[93] following the corresponding rice chromosomes and then mapped to the rice genome using MUMmer (v4.00rcl)[91]. Inversions were identified using SyRI (v1.5)[94] with parameters '-c -d -r -s–nosnp' with only these having no overlap with the breakpoint of chromosomal rearrangements detected above retained. The specific and shared inversions were determined using SURVIVOR (v1.0.7)[95] merge with parameters '0.4 1'.

We used a method based on phylogenetic patterns to identify putative homoeologous exchanges (HEs) between subgenomes[96] within polyploid bamboo genomes. Specially, we examined each individual gene tree to detect clusters of homoeologous copies with those from different subgenomes together as putative HEs. To achieve this, we selected rice with 11 bamboo genomes to infer orthogroups and phylogenetic trees using OrthoFinder (v2.5.2)[77], and subgenomes of WBs were treated as operational units in analysis.

## Gene retention evaluation

To assess gene retention patterns related to polyploidization, nine WB genomes and the combined two HB genomes (to make an artificial tetraploid genome for comparison with WBs) were aligned in CoGe's SynMap2 program with the LAST algorithm[97]. The maximum distance between two matches was set to 20 genes, and the minimum number of aligned pairs was set to 10 genes. Syntenic depth was calculated with 'Quota Align' with the ratio for bamboo to rice genes as 2:1 for combined HB and tetraploid genomes and 3:1 for hexaploids. Fractionation bias was then calculated using a window size of 100 genes, and only syntenic genes in the target genome were used for calculation.

## Inference of gene families and homoeologous groups

We selected five grass species (rice, sorghum, *Oropetium thomaeum* (phytozome V12), *Brachypodium distachyon* (Gramene V60) and *Triticum urartu* (http://gigadb.org/dataset/100050)), together with the 11 bamboo genomes, for inferring gene families and homoeologous groups. The gene family expansion and contraction analysis was performed using CAFÉ (v4.2.1)[98] with a random birth-and-death model. We also validiated the pattern of gene fractionation in subgenomes by mapping the short sequencing reads to the genome assembly by Bowtie2 (v2.3.4.1)[50] to compare the coverage of genes retained in single and two copies across subgenomes in tetraploids or in single, two and three copies across subgenomes in hexaploids. The microsynteny of the 1:1 (tetraploids)/1:1:1 (hexaploids) homoeologs of subgenomes was checked using MCScanX[99] within individual bamboo genomes, and those validated gene pairs/triads were used for analyses.

## Transcriptome analyses

The quality of RNA-seq reads was evaluated using FastQC (v0.11.8)[100], and raw reads were trimmed by Fastp (v0.20.1)[101]. Clean reads were aligned to genomes using HISAT2 (v2.1.0)[65] with duplicated aligned reads removed by SAMtools (v1.10)[102]. The remaining aligned reads were counted using a union-exon approach with StringTie[67] to get their gene set. The StringTie-HISAT2 approach[103] was used to correct the multi-mapping for a small portion of reads. Transcripts per kilobase million (TPM) fragments mapped were calculated for each gene by normalizing the read counts to both the length of the gene and the total number of mapped reads in the sample. Raw counts were normalized using the variance stabilizing transformation method (vst) in DESeq2 (v1.14.1)[104]. A hierarchical clustering analysis was used to ensure that the replicates clustered tightly to identify three outliers not clustered together with other replicate samples to be excluded. The expressed genes were counted requiring TPM ≥ 1 in at least two samples.

For PCA, TPM values for the expressed genes were transformed by ($\log_2$(TPM + 1)) and analyzed using the prcomp function in R v4.0.3 (https://www.r-project.org/). The neighbor-joining tree of all kinds of tissues sampled in *D. sinicus* was constructed by the ape (5.6-2) R package based on the expression matrix.

## Expression divergence between subgenomes

To determine expression patterns of homoeologs between subgenomes, we used the 1:1/1:1:1 gene pairs/triads identified above for analyses, and those from the mosaic chromosome of chr9 in TWBs were excluded. We also excluded *Rh. racemiflorum* for only with a few tissues for RNA samplings. We further identified 4,123 and 3,839 1:1 gene pairs shared by all three species of TWBs and NWBs, respectively, and 1,157 triads shared by three species of PWBs for analyses of expression divergence in each clade. PCA clustering was conducted as described above with the expression values averaged across biological replicates. Moreover, the $\log_2$((TPM C + 0.01)/(TPM D + 0.01)) and $\log_2$((TPM C + 0.01)/(TPM B + 0.01)) value of homoeologous pairs across five common tissues (vegetative leaf blade, vegetative leaf sheath, shoot, root, and rhizome) in TWBs and NWBs, respectively, and $\log_2$(((TPM A + 0.01/(TPM A + TPM B + TPM C + 0.01))/0.33), $\log_2$(((TPM B + 0.01/(TPM A + TPM B + TPM C + 0.01))/0.33) and $\log_2$(((TPM C + 0.01/(TPM A + TPM B + TPM C + 0.01))/0.33) in PWBs were used for clustering analysis by R function 'heatmap2'.

The homoeologous pairs/triads were clustered into 10 groups using the 'average method' based on the expression level and patterns of all components in the five common tissues noted above. We defined homoeologous genes from a pair/triad clustered into the same group as having a similar expression pattern and those into different groups as

shifted in expression patterns. Homoeologous pairs in the tetraploids with the same number of genes as in the hexaploids were randomly selected for clustering simulations.

## Expression bias between subgenomes

To measure the gene expression differences between 1:1 gene pairs in tetraploids, we performed differential expression analysis using the DESeq2 package (v1.14.1)[104]. Only genes with Benjamini-Hochberg-adjusted $P < 0.05$ and $\log_2$(fold change) ≥ 1 were retained.

The analysis of subgenome bias of expression is more complex in hexaploids, and we implemented three different analytic methods:

**(a) Differential expression**

As in tetraploids, we also identified genes differentially expressed between each pair of the three subgenomes (A versus B, A versus C and B versus C) in hexaploids.

**(b) Normalization of relative expression levels of the A, B and C subgenomes**

This analysis focused exclusively on the 1:1:1 gene triads in PWBs following Ramírez-González, et al. [32]. Briefly, we defined a triad as expressed when the sum of the A, B and C subgenome homoeologs had TPM > 0.5 and standardized the relative expression of each homoeolog across the triad. The ternary diagrams were plotted using the R package ggtern[105].

**(c) Definition of homoeologous expression bias categories**

The ideal normalized expression bias for the six categories was defined as in wheat[32]. We calculated the Euclidean distance (R function rdist) from the observed normalized expression of each triad to each of the six ideal categories. The shortest distance was used to assign the homoeolog expression bias category for each triad, and this was done for each tissue.

## Co-expression analysis and hub genes

The WGCNA R package (v1.69)[33] was used to build the co-expression network for *P. edulis*, *G. angustifolia*, *D. sinicus* and *Ra. guianensis*. To reduce the weight of highly expressed genes on correlation coefficients, we transformed TPM values by $\log_2$(TPM + 1), which compressed large values while preserving the relative magnitude of small values. The soft power threshold of 26, 10, 14 and 20 in *P. edulis*, *G. angustifolia*, *D. sinicus* and *Ra. guianensis*, respectively, was used as the first power to exceed a scale-free topology fit index of 0.9. A signed hybrid network was constructed blockwise in three blocks using the function blockwiseModules and a biweight mid-correlation 'bicor' with maxPOutliers = 0.05. The topographical overlap matrices were calculated by the blockwiseModules function using TOMType = 'unsigned', and the minimum module size was set to 30. Similar modules were merged by the parameter mergeCutHeight=0.15. Modules were tested for correlations with tissues using the cor() function. The significance of correlations was calculated using the function corPvalueStudent() and corrected for multiple testing by p.adjust()[106].

Hub genes within the module were identified using the function moduleEigengenes and signedKME (KME > 0.9). We took each gene in the module as a core and counted its 100 most associated genes based on the rank of Weight values in the co-expression network and calculated the frequencies of inter- and intra-subgenome interactions.

## Identifying new genes, PSGs and tissue-specific expressed genes

We followed the pipeline of Jin et al.[34], using the same 65 outgroups as they did, to date genes of the 11 bamboo genomes along the phylogenetic tree. The transcriptome age index (TAI) was calculated via the 'myTAI' R package (v0.9.3)[107,108] using the gene age and expression data from different tissues of *P. edulis* and *D. sinicus*, respectively.

To address the challenge of multiple gene copies in polyploids in identifying positively selected genes (PSGs), we used a subgenome-based approach (Supplementary Information). Positive

selection signals on genes along the common branch leading to the subgenome lineage of WBs were detected using the branch-site model by the Codeml program in the PAML package (v4.8)[109].

For tissue-specific expressed genes, we selected *D. sinicus* and *P. edulis* for analyses with the densest of RNA-seq samplings. Pairwise comparison between tissues were made by DESeq2 (v1.14.1)[104]. We further identified vegetative and reproductive stage-specific expressed genes of *Ra. guianensis*, *P. edulis*, *Rh. racemiflorum*, *B. amplexicaulis* and *D. sinicus* for analyses of nonsynonymous substitution (*Ka*) and synonymous substitution (*Ks*) values by KaKs-Calculator (v2.0)[110].

## Growth pattern of *D. sinicus* shoot

During the shooting season of *D. sinicus* in July and August 2020, we continuously measured the height of the whole shoot and the 9th, 10th and 11th internodes length of *D. sinicus* until the completion of their full elongation in Cangyuan County, Yunnan, China (Supplementary Information). We quantified the content of lignin, cellulose and hemicellulose in the 10th internode of *D. sinicus* shoot and performed anatomical observation of it at different stages during fast growth. The content of lignin in the middle internode of the mature shoot of *A. luodianensis*, *B. amplexicaulis*, *D. sinicus*, *H. calcarea* and *P. edulis* was also determined with at least 10 biological replicates. The content of lignin, cellulose and hemicellulose was measured by the acetyl bromide method[111] and modified dilute acid hydrolysis method[112], respectively.

## Identification of lignin genes and their expression

To investigate the molecular basis of the lignification process in bamboos, we identified the genes related to lignification in 11 bamboo species and five other grasses as above plus maize (*Zea mays*). The known genes in the lignin biosynthesis pathway (https://cellwall.genomics.purdue.edu) from *Arabidopsis thaliana* were used as seed sequence to identify their homologues in bamboos and the other grasses. BLAST hits with a percentage identity >35% and e-value < 1e-10 were kept for multiple sequence alignment by MAFFT v7.475 using default parameters[74]. Phylogenetic trees were built using IQ-TREE2 (v2.0.3)[113], and lignin-related genes in bamboos and other grasses were inferred. Identification of differentially expressed genes between four growth stages of *D. sinicus* was carried out and DEGs were grouped into clusters by using Short Time series Expression Miner (STEM) (v1.3.13)[114].

## Reporting summary

Further information on research design is available in the Nature Portfolio Reporting Summary linked to this article.

## Data availability

The 11 bamboo genome assemblies (GenBank numbers JAYEVB000000000, JAYEVC000000000, JAYEVD000000000, JAYEVE000000000, JAYEVF000000000, JAYEVG000000000, JAYEVH000000000, JAYEVI000000000, JAYEVJ000000000, JAYEVK000000000 and JAYGGG000000000), raw sequencing data and RNA-seq data are available at NCBI (accession: PRJNA948693). Genomes and annotations can be accessed at CoGe (https://genome-evolution.org/coge/NotebookView.pl?nid=3091) and our bamboo omics and systematics database (https://bamboo.genobank.org/). Source data are provided with this paper.

## Code availability

The custom codes included in this study are available at GitHub (https://github.com/yunlongliukm/BGSP). Codes are also archived at Zenodo (https://doi.org/10.5281/zenodo.10146649 (ref. 115)).

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

## Acknowledgements

We thank J.-Y. Hu and Y. Lu for inspiring discussion and comments and L.-M. Liu, K.-C. Qian, H. Wu and Y. Luo for help with sample collection. This work was supported by the Strategic Priority Research Program of Chinese Academy of Sciences (XDB31000000 to D.-Z.L.), the National Natural Science Foundation of China (32120103003 to D.-Z.L. and 31970355 to P.-F.M.), Leading Talents Program of Yunnan Province (2017HA014 to D.-Z.L.), Youth Innovation Promotion Association of CAS (Y201972 to P.-F.M.) and China Postdoctoral Science Foundation (2022T150664 to G.J.) and facilitated by the Germplasm Bank of Wild Species.

## Author contributions

D.-Z.L. conceived and designed the project. P.-F.M., Z.-H.G., and Y.-L.L. coordinated the project. C.G., P.-F.M., L.M. and Z.-H.G. collected and prepared the samples with assistance from L.-Z.N., Z.-C.X., Y.-J.W., Y.L, Y.Y. and X.-Y.Y. Y.-L.L., G.J, C.G., P.-F.M., Y.-Z.Y., L.M. and L.-Z.N. performed bioinformatics analyses and analyzed data with contributions from Y.-J.W., J.-X.L. and M.-Y.Z. L.G.C., E.A.K., D.E.S., J.L.B. and P.S.S. contributed valuable suggestions to analyses and interpretation of results. P.-F.M., D.-Z.L., C.G., G.J, Y-L.L., Y.-Z.Y. and L.M. wrote the paper with input from L.G.C., E.A.K., D.E.S., J.L.B. and P.S.S. All authors read and approved the paper.

## Competing interests

The authors declare no competing interests.

## Additional information

**Extended data** is available for this paper at https://doi.org/10.1038/s41588-024-01683-0.

**Correspondence and requests for materials** should be addressed to De-Zhu Li.

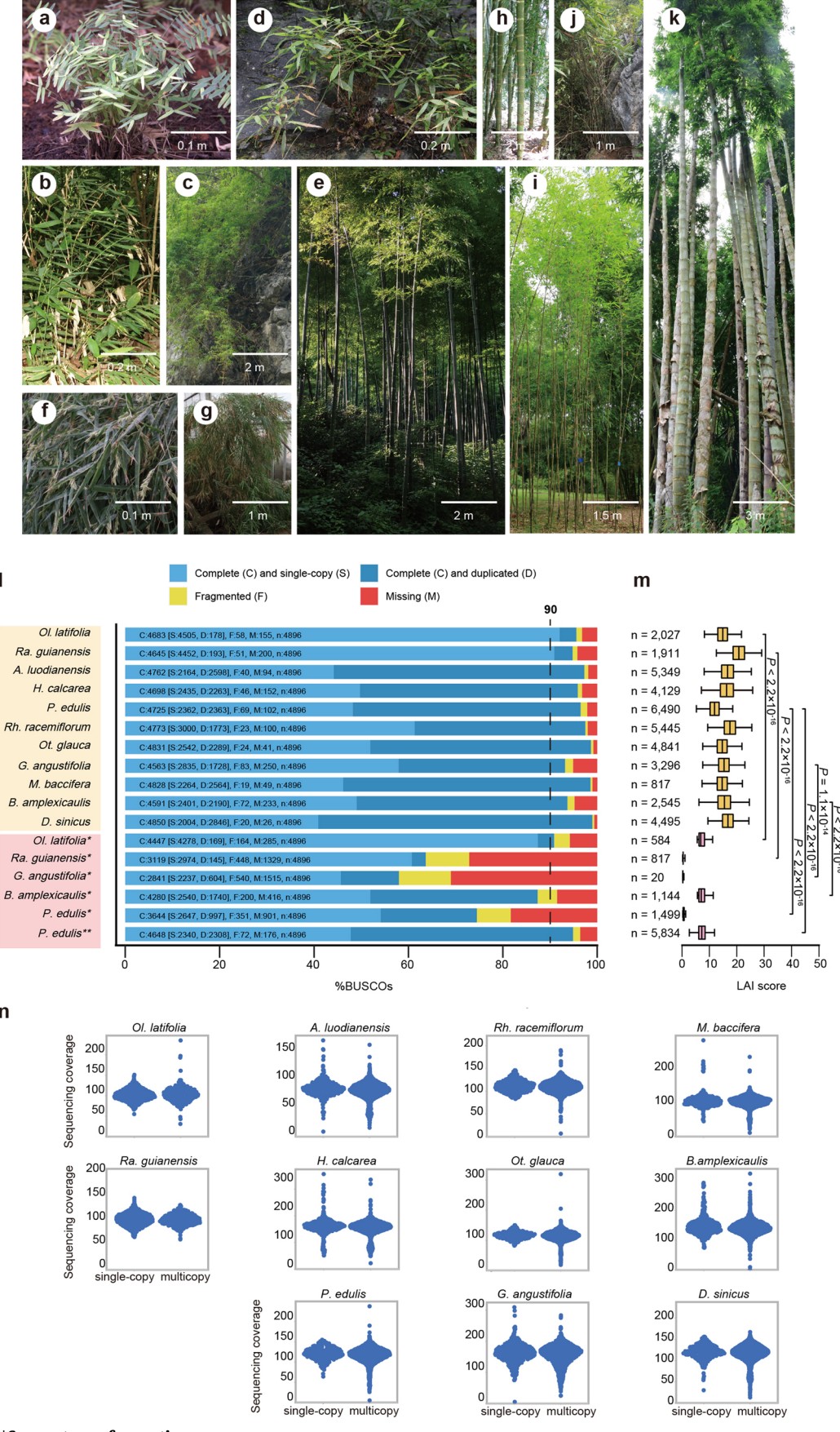

Extended Data Fig. 1 | See next page for caption.

**Extended Data Fig. 1 | Morphological features of 11 sequenced bamboo species and evaluation of quality of the genome assemblies. a**, *Raddia guianensis* (Rgu), scale bar = 0.1 m. **b**, *Olyra latifolia* (Ola), scale bar = 0.2 m. **c**, *Ampelocalamus luodianensis* (Alu), scale bar = 2 m. **d**, *Hsuehochloa calcarea* (Hca), scale bar = 0.2 m. **e**, *Phyllostachys edulis* (Ped), scale bar = 2 m. **f**, *Otatea glauca* (Ogl), scale bar = 0.1 m. **g**, *Rhipidocladum racemiflorum* (Rra), scale bar = 1 m. **h**, *Guadua angustifolia* (Gan), scale bar = 2 m. **i**, *Melocanna baccifera* (Mba), scale bar = 1.5 m. **j**, *Bonia amplexicaulis* (Bam), scale bar = 1 m. **k**, *Dendrocalamus sinicus* (Dsi), scale bar = 3 m. **l**, Completeness evaluation of annotated genes for the 11 bamboo genomes assessed using Benchmarking Universal Single-Copy Orthologs (BUSCO). Previous assemblies of five species[17,38,116] are indicated by '*' and '**'. **m**, Continuity of genome assembly assessed by LTR Assembly Index (LAI). Boxplots: centerline, median; box limits, first and third quartiles; whisker, 1.5x interquartile range; two-sided Wilcoxon rank-sum test. **n**, Assembly quality evaluation based on the sequencing coverage distributions of genes from single-copy and multicopy BUSCO orthogroups using calculate_AG in Mabs. In the 11 assemblies, the coverage distributions are nearly the same for genes between single-copy orthogroups and multicopy orthogroups.

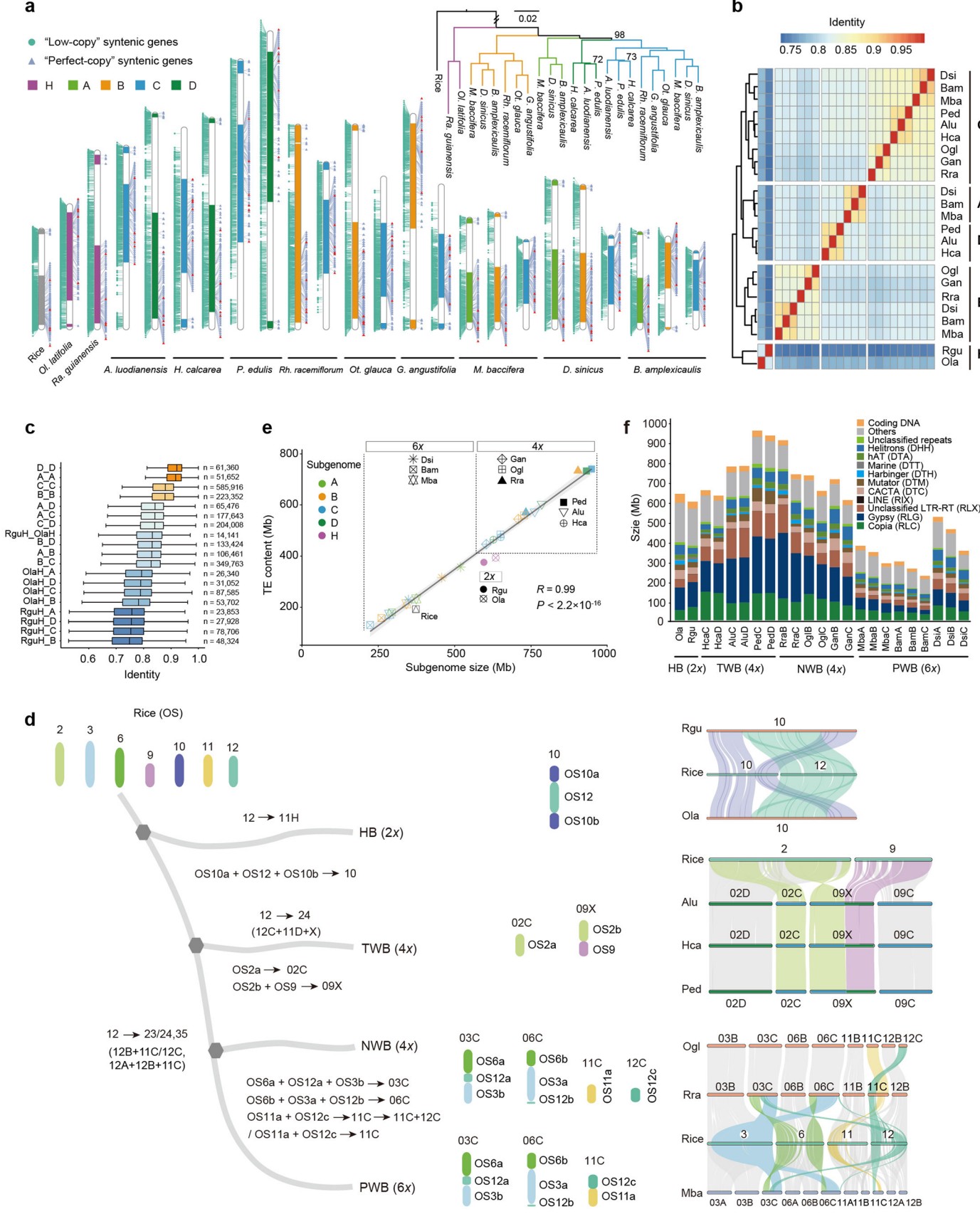

**Extended Data Fig. 2 | See next page for caption.**

**Extended Data Fig. 2 | Identification of subgenomes and reduction of chromosome numbers by rearrangement in major bamboo clades.**
**a**, Distribution of 'perfect-copy' (456) and 'low-copy' (13,891) syntenic genes along the 12 chromosomes of rice genome and 11 bamboo genomes, with chr1 as an example shown here. See Supplementary Fig. 6 for the remaining 11 chromosomes. The red triangles represent genes filtered by putative gene conversion or highly deviating from the ASTRAL species tree. Colored bands represent blocks in which 'perfect-copy' syntenic genes are clustered and different colors correspond to the identified subgenomes (H, A, B, C and D). The phylogenetic tree inferred by concatenated 'perfect-copy' syntenic genes from the longest block was shown on the upper right with only nodes supported by bootstrap values lower than 100% shown. **b**, **c**, The average sequence identity from all syntenic gene pairs between subgenomes are shown in the heat map (**b**)

and all sequence identity from specific subgenome pairs of different species are drawn in the boxplots (centerline, median; box limits, first and third quartiles; whisker, 1.5x interquartile range) (**c**). **d**, Construction process and synteny pattern of related chromosomes are shown: the chr10-chr12 nested chromosome fusion (NCF) found in herbaceous bamboo (HB), mosaic chromosome by fusion between chr9D and a large segment of chr2C in temperate woody bamboo (TWB), and fission and fusion of chr12 into chr3, chr6 and chr11 of the C subgenome in the tropical clades of neotropical woody bamboo (NWB) and paleotropical woody bamboo (PWB). **e**, Correlation between subgenome size and transposable element (TE) content. Pearson's correlation coefficient was computed, followed by a two-sided *t*-test to ascertain the significance of the relationship. **f**, Repeat content of bamboo subgenomes.

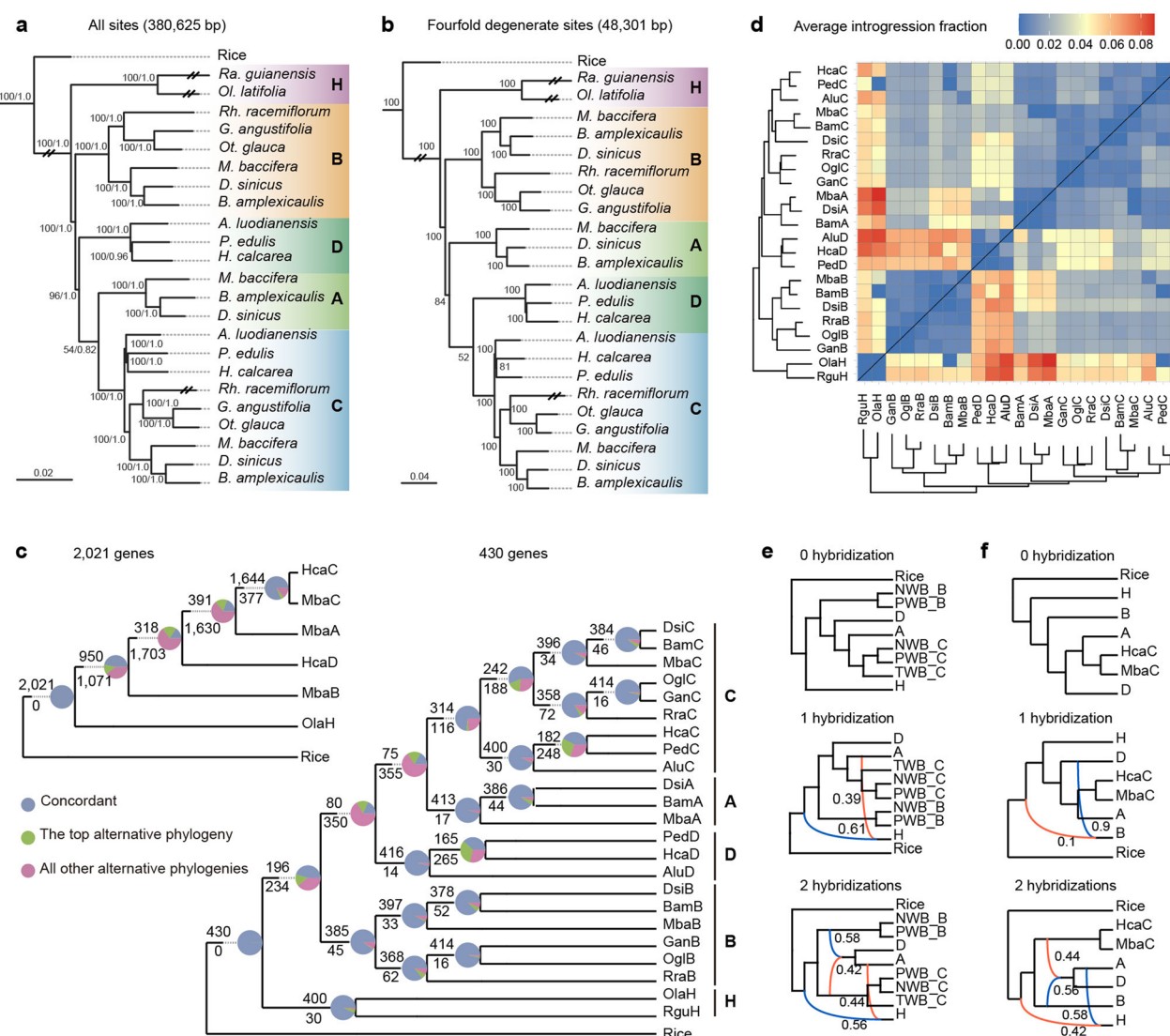

**g**

| Gene set | Observed topologies | | | | | |
|---|---|---|---|---|---|---|
| | **Monophyly of woody bamboos** | | **Plastid like** | | **H clustered with A** | |
| | Gene number | Proportion | Gene number | Proportion | Gene number | Proportion |
| 430 | 196 | 0.46 | 27 | 0.06 | 74 | 0.17 |
| 2,021 | 950 | 0.47 | 168 | 0.08 | 320 | 0.16 |
| | **H(B,C)** | | **B(H,C)** | | **C(H,B)** | |
| | Gene number | Proportion | Gene number | Proportion | Gene number | Proportion |
| 430 | 246 | 0.57 | 105 | 0.25 | 79 | 0.18 |
| 2,021 | 1,146 | 0.57 | 509 | 0.25 | 366 | 0.18 |
| | **B(A,C)** | | **C(A,B)** | | **A(B,C)** | |
| | Gene number | Proportion | Gene number | Proportion | Gene number | Proportion |
| 430 | 206 | 0.48 | 121 | 0.28 | 103 | 0.24 |
| 2,021 | 919 | 0.46 | 592 | 0.29 | 510 | 0.25 |
| | **B(D,C)** | | **C(D,B)** | | **D(B,C)** | |
| | Gene number | Proportion | Gene number | Proportion | Gene number | Proportion |
| 430 | 199 | 0.46 | 132 | 0.31 | 99 | 0.23 |
| 2,021 | 855 | 0.42 | 677 | 0.34 | 489 | 0.24 |

**Extended Data Fig. 3 | See next page for caption.**

**Extended Data Fig. 3 | Conflicting phylogenetic relationships and inferred hybridization/introgression events among the major subgenome lineages.**
**a**, **b**, Maximum likelihood phylogenetic trees inferred from all concatenated sites (**a**) and fourfold degenerate sites (**b**) of 430 'perfect-copy' syntenic genes. In **a**, the numbers on nodes represent bootstrap values inferred by RAxML/posterior probabilities inferred from ASTRAL based on 430 individual gene trees. **c**, Extensive topological discordance among individual gene trees by Phyparts analyses based on 430 and 2,021 'perfect-copy' syntenic genes. **d**, The heat map of average introgression fraction for each pair of subgenome inferred from QuIBL analyses based on the 430 gene data set. **e**, **f**, Two main hybridization scenarios revealed by Network analyses of 430 (**e**) and 2,021 (**f**) gene data sets. Solid (blue) and dashed (red) curved lines represent the major and minor edges that contribute to the hybrid descendants with the numbers indicating the inheritance probabilities of each parent. **g**, Distribution of observed conflicting topologies in 'perfect-copy' syntenic gene trees. Plastid like indicates the non-monophyly of woody bamboo as shown in Supplementary Fig. 13.

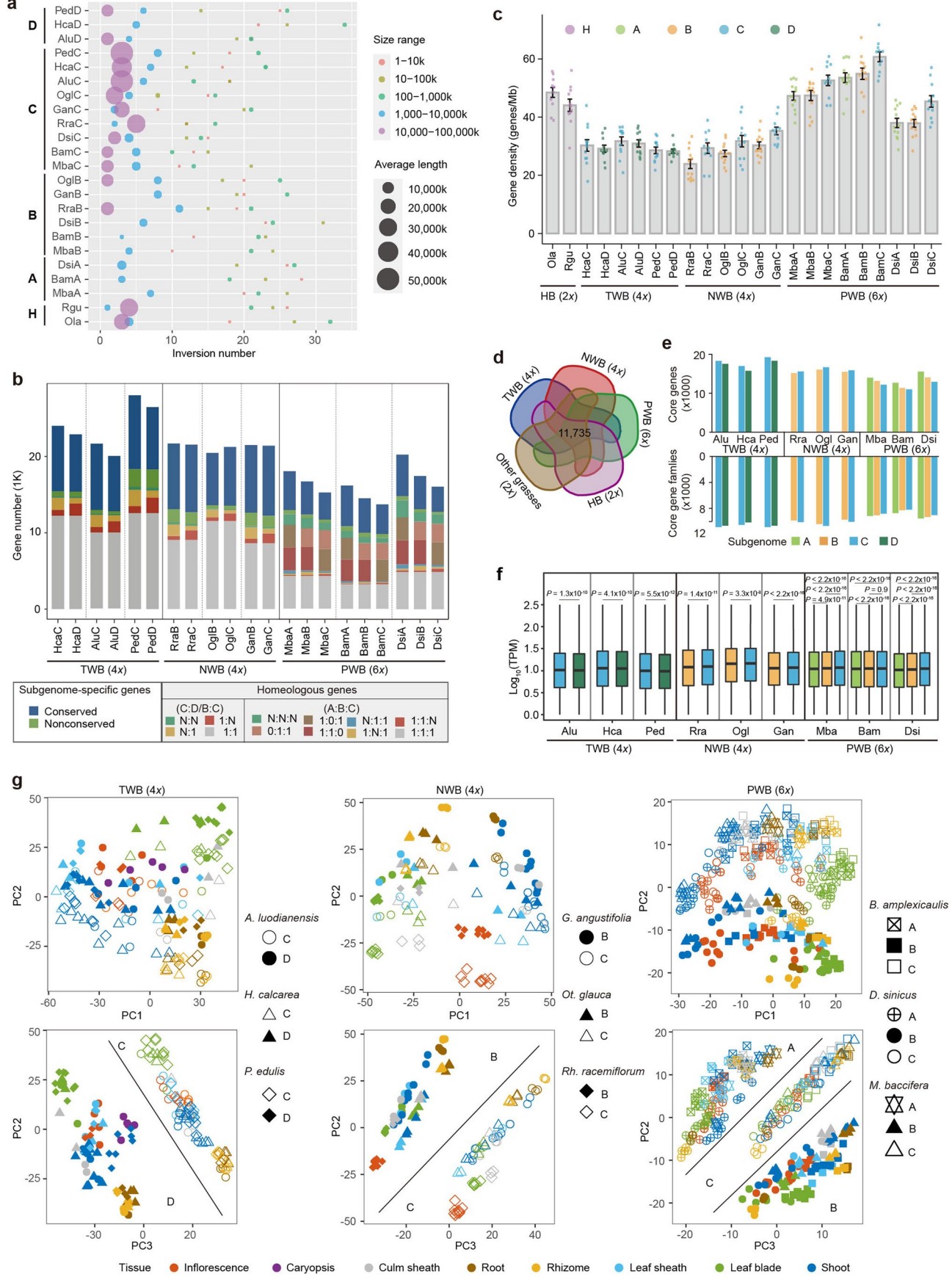

**Extended Data Fig. 4 | See next page for caption.**

**Extended Data Fig. 4 | Divergent evolution of subgenomes in bamboos.**
**a**, The distribution of number and size of detected inversions (>1 kb) in the bamboo genomes with the rice genome as reference. **b**, Groups of homoeologous genes in nine woody bamboo genomes. A total of 11 sequenced bamboo genomes and five other grass genomes of *Oropetium thomaeum*, *Sorghum bicolor*, *Oryza sativa*, *Triticum urartu* and *Brachypodium distachyon* were used for analyses. The subgenome-specific genes are those found only in one subgenome but not its counterpart(s) within the genome while with (conserved) or without (non-conserved) homoeologs in other genomes analyzed. **c**, Gene density of bamboo subgenomes calculated based on individual chromosomes ($n$ = 11 for Ola, Rgu, HcaD, AluD, PedD, RraC, GanC, MbaC, BamC and DsiC; $n$ = 12 for HcaC, AluC, PedC, RraB, OglB, OglC, GanB, MbaA, MbaB, BamA, BamB, DsiA, and DsiB) indicated by dots (error bar, mean ± s.e.m.). **d**, Venn diagram shows the number of core gene families of grasses shared by all the 16 genomes analyzed. **e**, The distribution of core gene families and genes in these families identified in **d** across the subgenomes of woody bamboos. **f**, Average transcript abundance across sampled tissues for accumulated expressed genes between subgenomes of woody bamboos ($n$ = 384,491 versus 367,998 in Alu, 363,971 versus 339,012 in Hca, 525,603 versus 498,493 in Ped, 277,692 versus 283,449 in Rra, 234,389 versus 249,521 in Ogl, 347,942 versus 368,737 in Gan, 280,660 versus 266,630 versus 255,050 in Mba, 779,750 versus 696,464 versus 691,827 in Bam, 1,206,253 versus 1,117,427 versus 1,071,291 in Dsi; two-sided Wilcoxon rank-sum test; boxplots: centerline, median; box limits, first and third quartiles; whisker, 1.5x interquartile range). **g**, Principal-component analysis (PCA) for similarity of expression of homoeologs across different tissues.

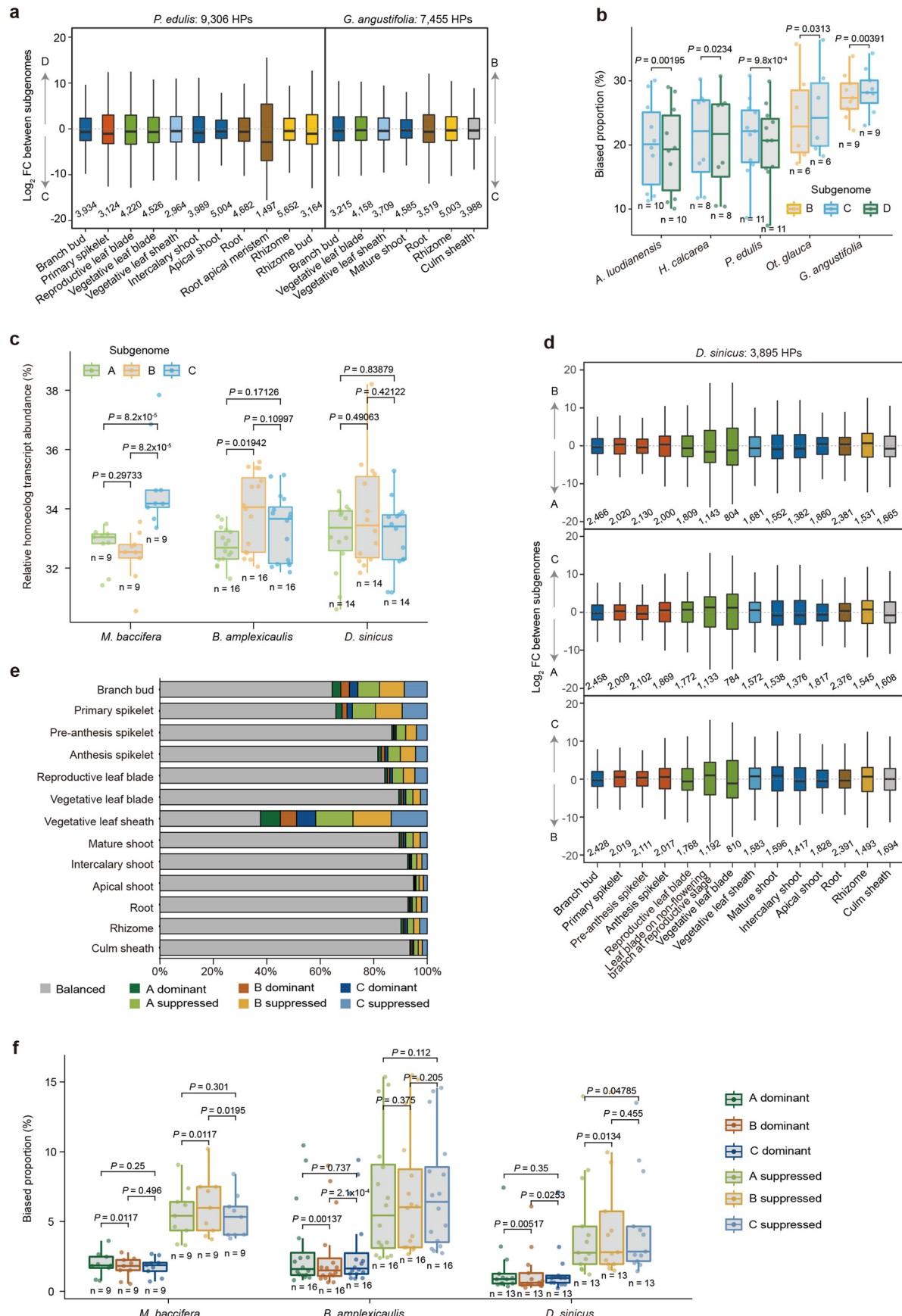

**Extended Data Fig. 5 | See next page for caption.**

**Extended Data Fig. 5 | Homoeolog expression bias among subgenomes of woody bamboos. a**, Boxplots of biased expression for homoeologous pairs across different tissues in representative tetraploid bamboos of *P. edulis* and *G. angustifolia*. **b**, Comparison of biased expression between subgenomes in tetraploid bamboos (two-sided Wilcoxon singed-rank test). **c**, Boxplots of relative expression abundance of A, B and C subgenomes in three hexaploid species (two-sided Wilcoxon rank-sum test). **d**, Boxplots of biased expression for homoeologous genes across different tissues in *D. sinicus* as representative of hexaploid bamboos. **e**, Proportion of triads in each category of homoeologous expression bias across 13 different tissues in *D. sinicus* as representative of hexaploid bamboos. **f**, Comparison of biased expression between subgenomes in hexaploid bamboos (two-sided Wilcoxon singed-rank test). The relative frequency of dominant and suppressed triads was compared among the three subgenomes. Boxplots in **a-d** and **f**: centerline, median; box limits, first and third quartiles; whisker, 1.5x interquartile range.

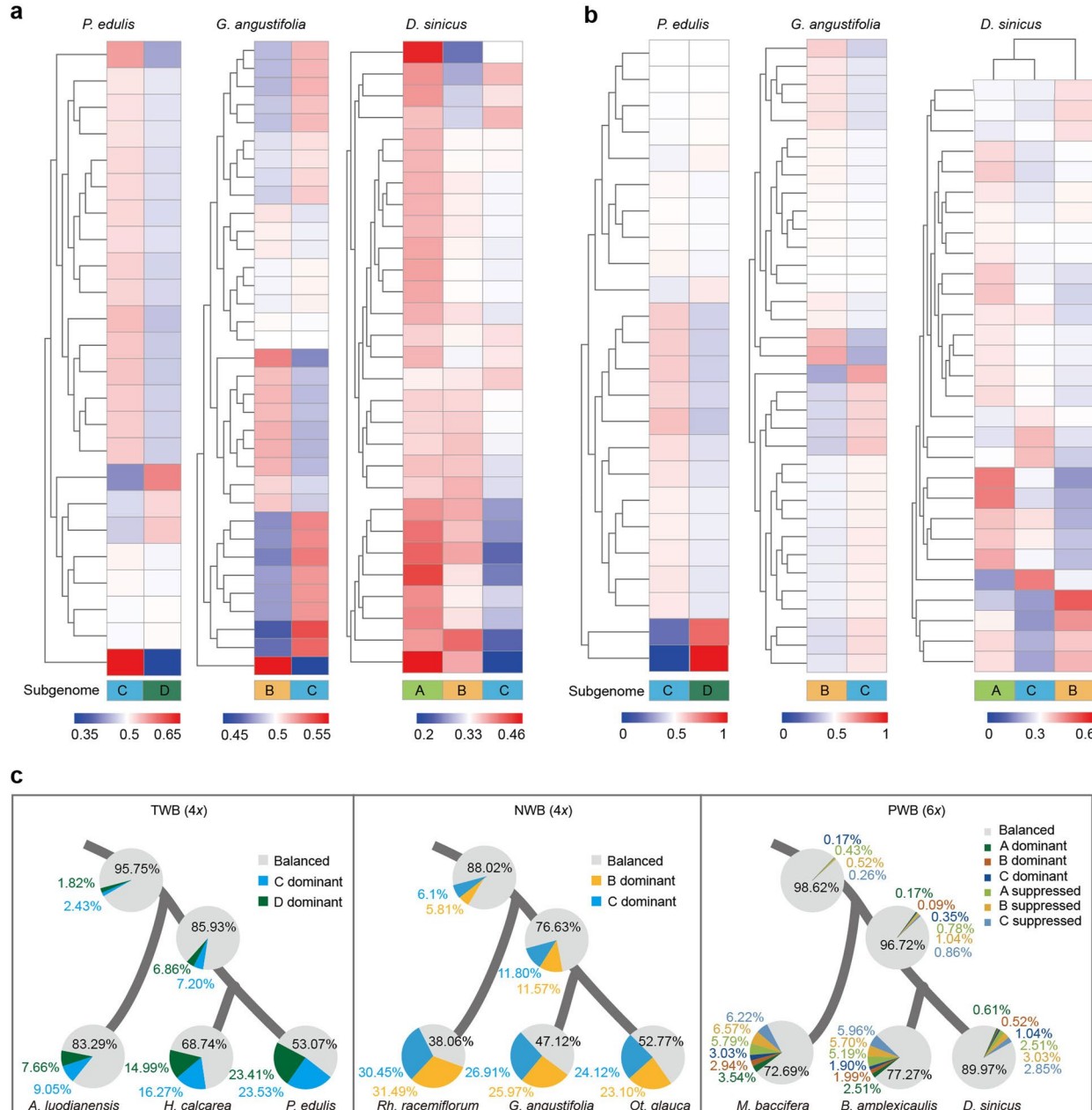

**Extended Data Fig. 6 | Biased subgenomes on gene expression and their origin. a**, **b**, Heatmap representation of WGCNA modules showing the percentage of co-expressed (**a**) and hub (**b**) genes from different subgenomes. **c**, Origin and evolution of subgenome expression bias in three woody bamboo clades. Subgenome bias of gene expression was estimated based on the vegetative leaf blade.

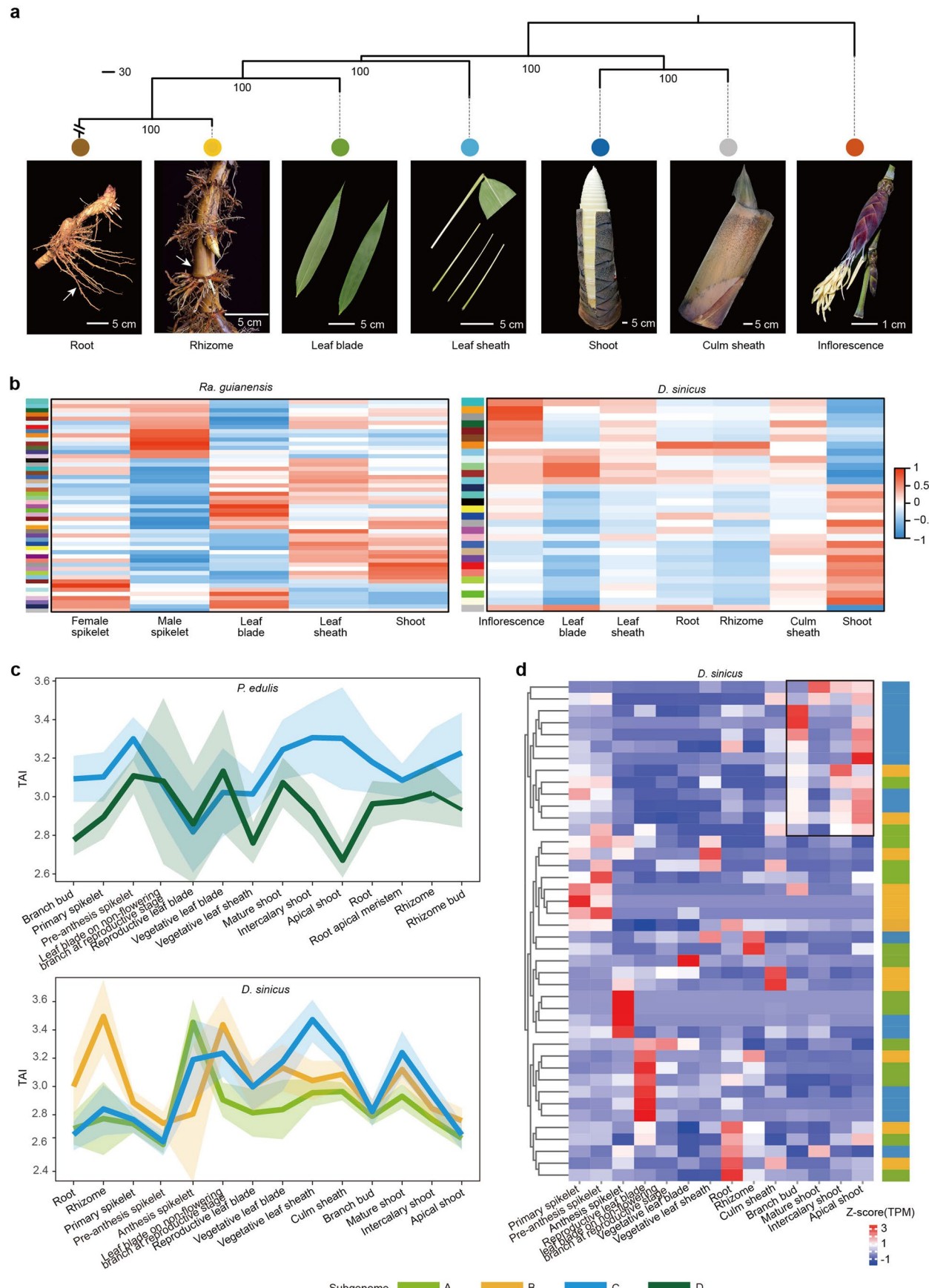

**a**

**b** *Ra. guianensis* *D. sinicus*

**c** *P. edulis* *D. sinicus*

**d** *D. sinicus*

Subgenome ▮ A ▮ B ▮ C ▮ D

**Extended Data Fig. 7 | See next page for caption.**

**Extended Data Fig. 7 | The evolution of gene expression in bamboo tissues.**
**a**, The neighbor-joining (NJ) tree of bamboo tissues based on transcriptome distances in *D. sinicus*. The number at nodes indicate bootstrap values estimated for 1,000 replicates. Scale bars are 1 cm for inflorescence and 5 cm for other tissues, respectively. **b**, Correlation between the module eigengene (kME; representative gene expression pattern) and the tissue in the WGCNA co-expression network in *Ra. guianensis* and *D. sinicus*. **c**, Transcriptome age index (TAI) across different tissues in *P. edulis* and *D. sinicus*. The shaded bands represent the standard deviation of TAI. **d**, Expression heat map of 42 new genes across different tissues in *D. sinicus*. The black box indicates those specifically expressed in the shoot.

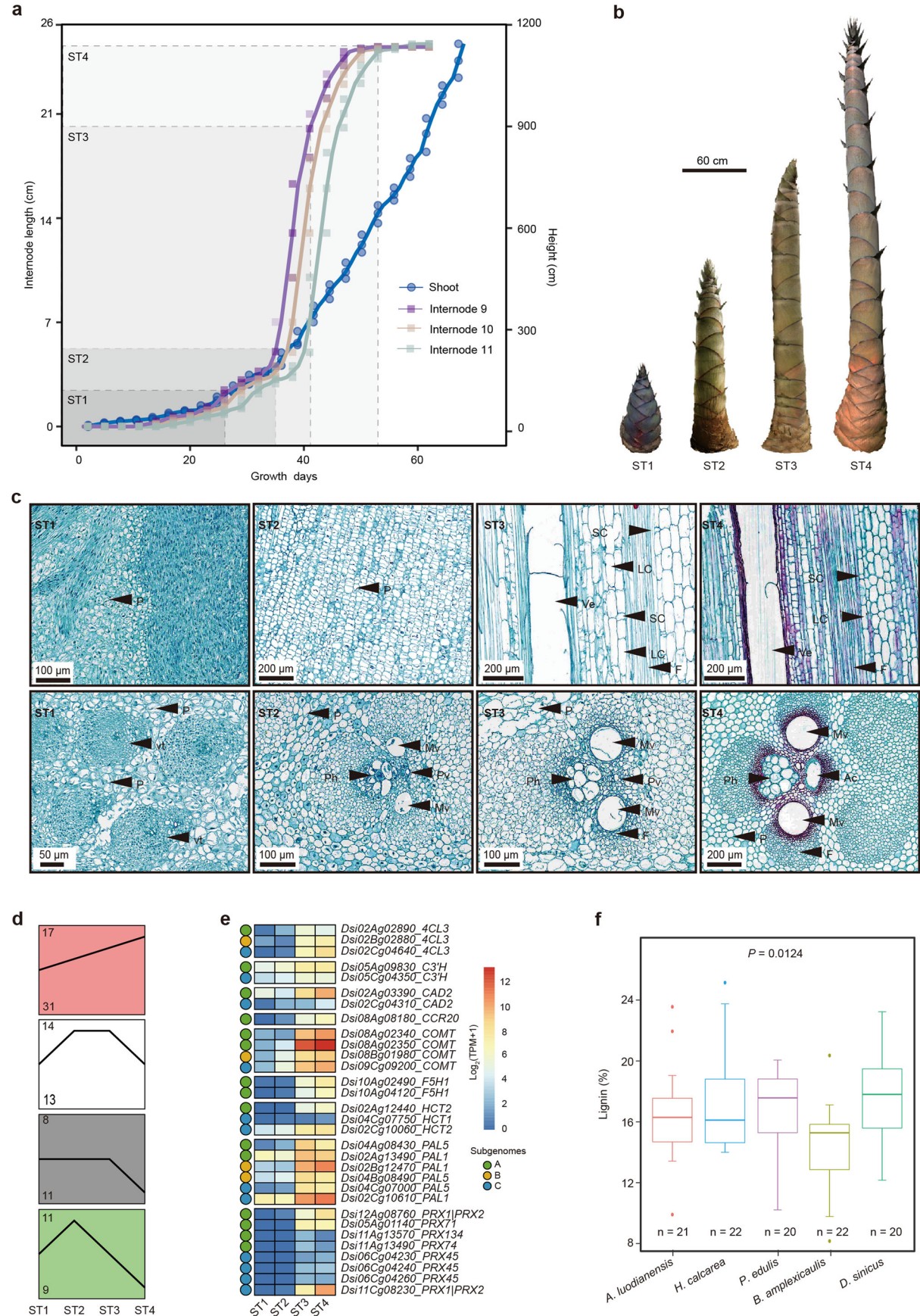

**Extended Data Fig. 8 | See next page for caption.**

**Extended Data Fig. 8 | Analyses of rapid growth and lignification of shoot in
*D. sinicus*. a**, The growth curves of the shoot (right label) and the 9th, 10th, and
11th internodes (left label) showing a pattern of 'slow-fast-slow' growth, which
could be divided into four stages from stage 1 (ST1) to stage 4 (ST4). **b**, The height
of *D. sinicus* shoot at four different stages. Scale bar = 60 cm. **c**, Micrographs of
longitudinal (scale bars = 100 μm for ST1 and 200 μm at ST2 to ST4, respectively)
and transverse (scale bars = 50 μm for ST1, 100 μm for ST1 and ST3, and 200 μm
for ST4) sections of the 10th internode at four different stages. The experiment
was independently repeated three times. P: parenchyma cells; Vt: vascular tissue;
Ph: phloem; Pv: protoxylem vessel; Mv: metaxylem vessel; F: fiber cells; LC: long
parenchyma cells; SC: short parenchyma cells; Ve: vessel; Ac: cavity formed by the
degradation of the protoxylem. **d**, The top four clusters of 114 genes enriched in
lignin biosynthesis in the KEGG pathway by the STEM software. Profile number
labeled on the upper left corner and the number of genes on the lower left corner.
Colored clusters are those having genes significantly enriched (Bonferroni
adjusted $P < 0.05$, the permutation test). **e**, The expression pattern of 31 genes
significantly positively correlated with the lignin content of *D. sinicus* shoot at
four stages of ST1 to ST4. **f**, Comparison of lignin content of developed shoots
among five woody bamboo species, with the highest level found in *D. sinicus*
(Kruskal-Wallis test; boxplots: centerline, median; box limits, first and third
quartiles; whisker, 1.5x interquartile range).

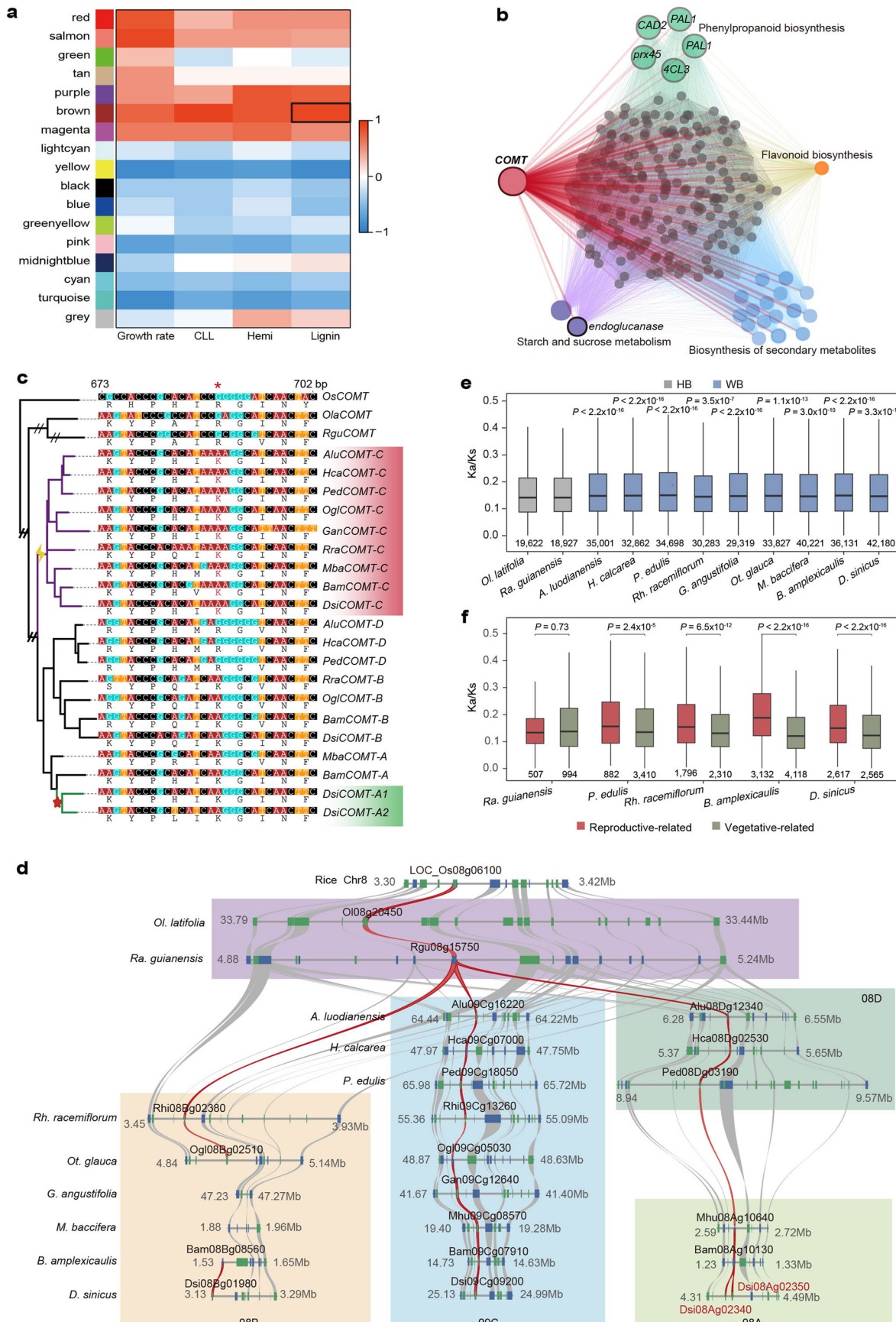

**Extended Data Fig. 9 | See next page for caption.**

**Extended Data Fig. 9 | The evolution and role of *COMT* in the lignin biosynthesis and comparison of molecular evolution between herbaceous and woody bamboos. a**, Correlation between gene modules and sampling traits of the *D. sinicus* shoot during rapid growth identified by WGCNA. The brown module containing *COMT* is significantly positively correlated with all four traits. Growth_rate: daily increments of the 10th internode of *D. sinicus* shoot; CLL: cellulose content; Hemi: hemicellulose content; Lignin: lignin content. **b**, *COMT* in the co-expression network correlated to the lignin content. **c**, Phylogenetic tree of *COMT* genes from 11 bamboo genomes. Detected positive selection is indicated by yellow lightning along the common ancestral branch leading to the C-subgenome copies. Sequence alignment shows specifically changed

sites in the C-subgenome copies. The tandem duplication in the A subgenome of *D. sinicus* is marked with a red star on the node. **d**, The syntenic relationships of *COMT* between bamboo and rice genomes. Syntenic genes are connected by curves with *COMT* indicated by red. **e**, Comparison *Ka/Ks* ratio between herbaceous bamboo (HB) and woody bamboo (WB). **f**, Comparison *Ka/Ks* ratio of specifically expressed genes in the leaf between reproductive stage (reproductive-related) and vegetative stage (vegetative-related). In **e** and **f**, the significance of difference was determined by two-sided Wilcoxon rank-sum test (boxplots: centerline, median; box limits, first and third quartiles; whisker, 1.5x interquartile range).

**Extended Data Table 1 | Summary statistics of 11 sequenced bamboo genomes**

| | HB (2x) | | TWB (4x) | | | NWB (4x) | | | PWB (6x) | | |
|---|---|---|---|---|---|---|---|---|---|---|---|
| | Ola | Rgu | Alu | Hca | Ped | Rra | Ogl | Gan | Mba | Bam | Dsi |
| Estimated genome size (Gb) | 0.68 | 0.63 | 1.69 | 1.32 | 1.98 | 1.72 | 1.51 | 1.58 | 1.07 | 0.87 | 1.4 |
| Heterozygosity (%) | 0.04 | 0.08 | 0.37 | 0.6 | 0.13 | 0.12 | 0.06 | 0.76 | 0.22 | 0.7 | 0.2 |
| Short reads (Gb) | 62.06 | 61.7 | 146.58 | 123.72 | 159.63 | 121.32 | 90.84 | 125.75 | 77.75 | 68.06* | 109.40 |
| Depth | 91.26x | 97.94x | 86.73x | 93.72x | 80.62x | 70.53x | 60.16x | 79.59x | 72.54x | 72.66x | 78.14x |
| Nanopore long reads (Gb) | 73.35 | 71.46 | 171.05 | 215.43 | 230.55 | 175.46 | 160.38 | 257.42 | 111.8 | 120.01 | 170.98 |
| Depth | 107.87x | 113.43x | 101.21x | 163.2x | 116.44x | 102.01x | 106.21x | 162.92x | 104.49x | 137.94x | 122.13x |
| Hi-C data (Gb) | 46.06 | 82.4 | 172.86 | 148.7 | 418.45 | 121.14 | 158.1 | 164.66 | 154.35 | 55.96 | 160.75 |
| Number of contigs | 114 | 917 | 1,389 | 920 | 1,944 | 305 | 1,439 | 3,619 | 818 | 1,557 | 1,048 |
| N50 of contigs (Mb) | 13.67 | 3.35 | 2.87 | 2.2 | 2.52 | 17.53 | 1.8 | 1.26 | 6.41 | 2.7 | 6.39 |
| Number of scaffolds | 43 | 605 | 574 | 116 | 954 | 135 | 484 | 2103 | 443 | 1121 | 557 |
| N50 of scaffolds (Mb) | 57.04 | 56.58 | 65.33 | 54.52 | 79.77 | 70.24 | 56.75 | 54.95 | 28.11 | 22.77 | 36.31 |
| Assembly size (Mb) | 639.19 | 612.6 | 1,659.99 | 1,298.05 | 2,036.37 | 1,711.78 | 1,448.63 | 1,545.93 | 1,077.07 | 850.46 | 1,439.8 |
| GC content (%) | 44.67 | 46.7 | 43.94 | 43.75 | 44.04 | 44.47 | 43.21 | 43.5 | 43.19 | 43.4 | 44.42 |
| Pseudo-chromosomes | 11 | 11 | 24 | 24 | 24 | 24 | 24 | 23 | 35 | 35 | 35 |
| Sequence in pseudomolecules (Mb) | 634.43 | 595.67 | 1,547.28 | 1,277.96 | 1,879.99 | 1,676.0 | 1,312.05 | 1,312.05 | 1,002.79 | 778.39 | 1,326.08 |
| Number of protein-coding genes | 31,189 | 27,496 | 48,870 | 41,907 | 58,664 | 44,113 | 43,141 | 47,971 | 51,908 | 47,213 | 56,847 |
| Repeats (%) | 61.9 | 62.95 | 76.61 | 73.39 | 78.56 | 79.93 | 76.16 | 77.21 | 62.62 | 61.3 | 68.51 |

Note: The species name of 11 bamboos was abbreviated by three letters: Ola, *Olyra latifolia*; Rgu, *Raddia guianensis*; Alu, *Ampelocalamus luodianensis*; Hca, *Hsuehochloa calcarea*; Ped, *Phyllostachys edulis*; Rra, *Rhipidocladum racemiflorum*; Ogl, *Otatea glauca*; Gan, *Guadua angustifolia*; Mba, *Melocanna baccifera*; Bam, *Bonia amplexicaulis*; Dsi, *Dendrocalamus sinicus*. HB, herbaceous bamboo; TWB, temperate woody bamboo; NWB, neotropical woody bamboo; PWB, paleo-tropical woody bamboo. *The sequencing data was from Guo et al., (2019)[17].

Summary statistics of 11 sequenced bamboo genomes.

# Reporting Summary

## Statistics

For all statistical analyses, confirm that the following items are present in the figure legend, table legend, main text, or Methods section.

| n/a | Confirmed | |
|---|---|---|
| ☐ | ☒ | The exact sample size (*n*) for each experimental group/condition, given as a discrete number and unit of measurement |
| ☐ | ☒ | A statement on whether measurements were taken from distinct samples or whether the same sample was measured repeatedly |
| ☐ | ☒ | The statistical test(s) used AND whether they are one- or two-sided <br> *Only common tests should be described solely by name; describe more complex techniques in the Methods section.* |
| ☒ | ☐ | A description of all covariates tested |
| ☐ | ☒ | A description of any assumptions or corrections, such as tests of normality and adjustment for multiple comparisons |
| ☐ | ☒ | A full description of the statistical parameters including central tendency (e.g. means) or other basic estimates (e.g. regression coefficient) AND variation (e.g. standard deviation) or associated estimates of uncertainty (e.g. confidence intervals) |
| ☐ | ☒ | For null hypothesis testing, the test statistic (e.g. *F*, *t*, *r*) with confidence intervals, effect sizes, degrees of freedom and *P* value noted <br> *Give P values as exact values whenever suitable.* |
| ☒ | ☐ | For Bayesian analysis, information on the choice of priors and Markov chain Monte Carlo settings |
| ☒ | ☐ | For hierarchical and complex designs, identification of the appropriate level for tests and full reporting of outcomes |
| ☒ | ☐ | Estimates of effect sizes (e.g. Cohen's *d*, Pearson's *r*), indicating how they were calculated |

*Our web collection on statistics for biologists contains articles on many of the points above.*

## Software and code

Policy information about availability of computer code

| Data collection | No software was used for data collection |
|---|---|
| Data analysis | Genome size and heterozygosity estimation: GenomeScope (https://github.com/schatzlab/genomescope); De novo genome assembly: CANU (v1.7), SMARTdenovo (v1.0.0) (https://github.com/ruanjue/smartdenovo), NextDenovo (v2.3.1) (https://github.com/Nextomics/NextDenovo), Racon (v1.4.21), Nextpolish (v1.3.0), Pilon (v1.23), BWA (v0.7.10-r789), Bowtie2 (v2.3.2), HiC-Pro (v.2.8.1), LACHESIS (https://github.com/shendurelab/LACHESIS); Assembly quality evaluation: BWA (v0.7.10-r789), LTR_retriever (v2.6), Mabs (v2.19) (https://github.com/shelkmike/Mabs); TE annotations: EDTA (v1.8.5), LTR_Finder (v1.07), Generic Repeat Finder (v1.0), TIR-Learner (v1.19), HelitronScanner (v1.1), RepeatModeler (v2.0.1) (https://github.com/Dfam-consortium/RepeatModeler); Gene annotations: Genscan, Augustus (v2.4), GlimmerHMM (v3.0.4), GeneID (v1.4), SNAP (v2006.07.28), GeMoMa (v1.3.1), HISAT (v2.0.4), Stringtie (v1.2.3), TransDecoder (v2.0) (https://github.com/TransDecoder/TransDecoder), GeneMarkS-T (v5.1), EvidenceModeler (v1.1.1), PASA (v2.0.2), BUSCO (v4.0.6); Methylation analysis: Bismark (v0.21.0), ViewBS (v0.1.9) (https://github.com/xie186/ViewBS); Subgenome identification: jcvi (v1.1.17), MAFFT (v7.471), PAL2NAL v14 (https://www.bork.embl.de/pal2nal/), RAxML (v8.2.12), OrthoFinder (v2.3.12), Identity (v1.0) (https://github.com/BioinformaticsToolsmith/Identity); Phylogenetic analysis and divergence time estimation: Newick utilities (v1.6.0) (https://github.com/tjunier/newick_utils), ASTRAL (v5.6.3), ipyrad (v0.9.74) (https://ipyrad.readthedocs.io/en/master/), MEGA-X, phyparts (v0.0.1) (https://bitbucket.org/blackrim/phyparts/src/master/), GetOrganelle (v1.7.1), MAFFT (v7.471), trimAl (v1.4), RAxML (v8.2.12), MCMCTREE in the PAML (v4.9); ILS and hybridization analysis: PhyloNet (v3.8.0), QuIBL (https://github.com/miriammiyagi/QuIBL), HyDe (v0.4.3) (https://github.com/pblischak/HyDe); Ancestral karyotype reconstruction: MCScanX (https://github.com/wyp1125/MCScanX), IAGS (https://github.com/xjtu-omics/IAGS); Identification of chromosomal rearrangements and inversions: jcvi (v1.1.17), MUMmer (v4.00rcl), bedtools (v2.30.0), EMBOSS-6.6.0, SyRI (v1.5), SURVIVOR (v1.0.7), OrthoFinder (v2.5.2); Gene retention evaluation: CoGe's SynMap2 (https://genomevolution.org/coge/); Inference of gene families and homoeologous groups: CAFÉ (v4.2.1) (https://github.com/hahnlab/CAFE.git), DIAMOND (v2.1.8) (https://github.com/bbuchfink/diamond), OrthoFinder (v2.5.2), Bowtie2 (v2.3.4.1), MCScanX (https://github.com/wyp1125/MCScanX), r8s (v1.8.1), (http:// |

ceiba.biosci.arizona.edu/r8s/r8s1.81.tar.gz), DupGen_finder (https://github.com/qiao-xin/DupGen_finder); Transcriptome analyses: FastQC (v0.11.8), Fastp (0.20.1), HISAT2 (v2.1.0), SAMtools (v1.10), StringTie (v1.3.4d) (http://ccb.jhu.edu/software/stringtie), DESeq2 (v1.14.1), R (v4.0.3), tispec R-package (https://rdrr.io/github/roonysgalbi/tispec); Expression divergence between subgenomes: R (v4.1.2); Expression bias between subgenomes: DESeq2 (v1.14.1), Co-expression analysis and hub genes: WGCNA (v1.69) (https://horvath.genetics.ucla.edu/html/CoexpressionNetwork/Rpackages/WGCNA/); Identifying new genes, PSGs and tissue-specific expressed genes: "myTAI" R package (v0.9.3) (https://github.com/drostlab/myTAI), PAML package (v4.8), DESeq2 (v1.14.1), KaKs-Calculator (v2.0), PAL2NAL v14, OrthoFinder (v2.5.2), MAFFT (v7.475), ParaAT (v2.0); Identification of lignin genes and their expression: OrthoFinder (v2.5.2), MAFFT (v7.475), IQ-TREE2 (v2.0.3), DESeq2 (v1.14.1), STEM (v1.3.13); Reference for all software have been described in the Methods. The custom codes included in this study are available at GitHub (https://github.com/yunlongliukm/BGSP). Codes are also archived at Zenodo (https://doi.org/10.5281/zenodo.10146649).

For manuscripts utilizing custom algorithms or software that are central to the research but not yet described in published literature, software must be made available to editors and reviewers. We strongly encourage code deposition in a community repository (e.g. GitHub). See the Nature Portfolio guidelines for submitting code & software for further information.

# Data

Policy information about availability of data

All manuscripts must include a data availability statement. This statement should provide the following information, where applicable:
- Accession codes, unique identifiers, or web links for publicly available datasets
- A description of any restrictions on data availability
- For clinical datasets or third party data, please ensure that the statement adheres to our policy

The 11 bamboo genome assemblies (GenBank numbers: JAYEVB000000000, JAYEVC000000000, JAYEVD000000000, JAYEVE000000000, JAYEVF000000000, JAYEVG000000000, JAYEVH000000000, JAYEVI000000000, JAYEVJ000000000, JAYEVK000000000 and JAYGGG000000000), raw sequencing data and RNA-seq data are available at NCBI (accession: PRJNA948693). Genomes and annotations can be accessed at CoGe (https://genomevolution.org/coge/NotebookView.pl?nid=3091) and the bamboo genomic resource website (http://bamboo.genobank.org/). Functional annotation of the genomes used the Poales_odb10 database (https://busco-data.ezlab.org/v5/data/lineages/poales_odb10.2020-08-05.tar.gz). Gene family inferences used the KEGG (https://www.genome.jp/kegg/kegg2.html) and GO (https://geneontology.org/) databases.

# Human research participants

Policy information about studies involving human research participants and Sex and Gender in Research.

| Reporting on sex and gender | N/A |
|---|---|
| Population characteristics | N/A |
| Recruitment | N/A |
| Ethics oversight | N/A |

Note that full information on the approval of the study protocol must also be provided in the manuscript.

# Field-specific reporting

Please select the one below that is the best fit for your research. If you are not sure, read the appropriate sections before making your selection.

☒ Life sciences    ☐ Behavioural & social sciences    ☐ Ecological, evolutionary & environmental sciences

For a reference copy of the document with all sections, see nature.com/documents/nr-reporting-summary-flat.pdf

# Life sciences study design

All studies must disclose on these points even when the disclosure is negative.

| Sample size | No statistical methods were required to establish sample size for this study. To cover different ploidal levels and genome diversity, 11 representative bamboo species were chosen for genome sequencing. Four bamboo species (Raddia guianensis, Phyllostachys edulis, Guadua angustifolia, and Dendrocalamus sinicus) representing herbaceous bamboos (HBs), temperate woody bamboos (TWBs), neotropical woody bamboos (NWBs) and paleotropical woody bamboos (PWBs) for whole-genome bisulfite sequencing (WGBS). A total of 476 transcriptome samples representing different tissues at various developmental stages across the 11 sequenced bamboos were sampled for RNA extraction and transcriptome sequencing, mostly with three biological replications per tissue per species. |
|---|---|
| Data exclusions | Raw sequence data were quality filtered as described in the manuscript. In the phylogenetic analysis, we removed 26 genes from the 456 "perfect-copy" syntenic gene data set and 654 genes from the 2675 "perfect-copy" syntenic gene data set based on the criteria as described in the Methods, respectively. |
| Replication | At least two biological replicates and three for the most were collected for each tissue type of RNA-seq. Two biological replicates were |

| Replication | collected for each leaf tissue sample for whole-genome bisulfite sequencing (WGBS). The experiment of anatomical observation of shoot in Dendrocalamus sinicus was repeated independently three times. Bootstrapping for phylogenetic analyses based on "perfect-copy" syntenic genes from 11 bamboo species and rice genome were replicated 200 times, while bootstrapping for phylogenetic analyses based on plastid genome sequences were replicated 1000 times. Two parallel runs were performed in the analyses of divergence time estimation. All attempts at replication were successful. |

| Randomization | Genomic analyses were conducted in a non-randomized order as we do not expect batch variations. |

| Blinding | Group allocation was not relevant to this study, so blinding was not necessary. |

# Reporting for specific materials, systems and methods

We require information from authors about some types of materials, experimental systems and methods used in many studies. Here, indicate whether each material, system or method listed is relevant to your study. If you are not sure if a list item applies to your research, read the appropriate section before selecting a response.

## Materials & experimental systems

| n/a | Involved in the study |
|---|---|
| ☒ | Antibodies |
| ☒ | Eukaryotic cell lines |
| ☒ | Palaeontology and archaeology |
| ☒ | Animals and other organisms |
| ☒ | Clinical data |
| ☒ | Dual use research of concern |

## Methods

| n/a | Involved in the study |
|---|---|
| ☒ | ChIP-seq |
| ☒ | Flow cytometry |
| ☒ | MRI-based neuroimaging |

