## [Peer Review File · Nature Genetics]

Peer Review Information

Manuscript Title: Genome assemblies of 11 bamboo species highlight diversification induced by dynamic subgenome dominance

Corresponding author name(s): Professor De-Zhu Li

Reviewer Comments & Decisions:

Decision Letter, initial version:

1st May 2023

Dear Professor Li,

Your Article, "Subgenome dominance induced diversification in the world's largest grasses" has now been seen by 2 referees. You will see from their comments copied below that while they find your work of considerable potential interest, they have raised quite substantial concerns that must be addressed. In light of these comments, we cannot accept the manuscript for publication, but would be very interested in considering a substantially revised version that fully addresses these serious concerns.

We hope you will find the referees' comments useful as you decide how to proceed. If you wish to submit a substantially revised manuscript, please bear in mind that we will be reluctant to approach the referees again in the absence of major revisions.

To guide the scope of the revisions, the editors discuss the referee reports in detail within the team with a view to identifying key priorities that should be addressed in revision. In this case, we think both referees have provided constructive reviews aimed at strengthening the analyses and improving the data quality. Importantly, Reviewer #2 notes that the initial quality checks of genome assemblies are lacking, which may affect the rest of the results. We particularly ask that you perform additional quality checks and ensure that the assembly's quality meets standards (Reviewer #2), include additional evidence to better support some of the conclusions (Reviewer #1), and address all referee comments as thoroughly as possible with appropriate revisions. We hope that you will find the prioritized set of referee points to be useful when revising your study.

If you choose to revise your manuscript taking into account all reviewer and editor comments, please highlight all changes in the manuscript text file. At this stage we will need you to upload a copy of the manuscript in MS Word .docx or similar editable format.

*2) If you have not done so already please begin to revise your manuscript so that it conforms to our Article format instructions, available [here](http://www.nature.com/ng/authors/article_types/index.html). Refer also to any guidelines provided in this letter.

[redacted]

If you wish to submit a suitably revised manuscript we would hope to receive it within 6 months. If you cannot send it within this time, please let us know. We will be happy to consider your revision so long as nothing similar has been accepted for publication at Nature Genetics or published elsewhere. Should your manuscript be substantially delayed without notifying us in advance and your article is eventually published, the received date would be that of the revised, not the original, version.

Nature Genetics is committed to improving transparency in authorship. As part of our efforts in this direction, we are now requesting that all authors identified as 'corresponding author' on published papers create and link their Open Researcher and Contributor Identifier (ORCID) with their account on the Manuscript Tracking System (MTS), prior to acceptance. ORCID helps the scientific community achieve unambiguous attribution of all scholarly contributions. You can create and link your ORCID from the home page of the MTS by clicking on 'Modify my Springer Nature account'. For more information please visit please visit

<http://www.springernature.com/orcid>>www.springernature.com/orcid.

Thank you for the opportunity to review your work.

Sincerely,
Wei

Wei Li, PhD
Senior Editor
Nature Genetics
New York, NY 10004, USA
www.nature.com/ng

Reviewers' Comments:

Reviewer #1:

Remarks to the Author:

The manuscript integrates extensive phylogenomics, comparative transcriptomic analyses to investigate the impact of polyploidization on species diversification using an extraordinary plant system, the polyploid bamboos. The study found that bamboo subgenomes exhibit remarkable karyotype stability, with parallel subgenome dominance in the two tetraploid clades and a gradual shift of dominance in the hexaploid clade. Furthermore, they found that polyploidy and subgenome dominance explain the genetic basis of biological traits of woody bamboos. This study provides valuable insights into the polyploidization and evolutionary history in the remarkable polyploid bamboo system. Here I have some questions about the data interpretation, however, not undermining the overall merit of the study.

Major concerns:

- 1) There are several possible reasons for the extensive topological discordance observed among coalescent-based trees. One plausible explanation could be the insufficient inference of introgression events among different genera and species. As the authors did not clarify how they addressed introgression events in their data, it is difficult to assess the extent to which such events may have affected their results.
- 2) The mosaic chromosome of Chr09D in TWB groups displays a distinct fusion pattern between chr09D and chr2C. However, the authors did not elaborate on this point, leaving room for further interpretation. One possible explanation is that homoeologous exchange, which is common among polyploid species, may have caused this mix of autopolyploid genome fragments in the allopolyploid genome. It would be interesting to know if the authors have identified smaller fragments showing a similar pattern in other polyploid genomes.
- 3) The authors of this study conducted extensive research on polyploid genome evolution. However, their findings regarding the influence of polyploidy on the diversification of specific traits are relatively weak. For instance, it is unclear whether the authors found evidence to support the dominance of the C subgenome in gene family size, novel genes, and PSGs related to woody culms and flowering.

Therefore, the authors should provide additional evidence to strengthen their argument regarding the impact of subgenome dominance on the diversification of specific traits.

4) Reproductive research. As the authors have conducted extensive in-depth analyses on genome and subgenome evolution, it would be valuable for the scientific community if they make their analysis pipeline and code publicly available. This would enable other researchers to reproduce and expand upon their work in future studies.

Minor:

1) The title. Since the research examined subgenome dominance in the remarkable polyploid bamboo system, it might be better to modify the title to include the "bamboo" instead of only "grasses".

2) Line 142. "...the set of 456 genes 26". It is unclear for the meaning of 2 numbers.

3) Line 194. "6 and 11 in the C subgenome" – unclear use of the numbers.

4) Line 381-383. How many PSGs are from C subgenomes. Do they show any enrichment for C subgenome?

Reviewer #2:

Remarks to the Author:

The submitted manuscript represents a great effort in genome assembly and analysis of subgenome dominance in the bamboo genus. This potentially very strong study has several issues. First, it is very hard to digest for a reviewer and future general reader. The authors need to think about how to present the data and results in a lighter way. Second, I feel that initial quality checks of genome assemblies are lacking, which may affect the rest of the results.

The topic of subgenome dominance is very interesting with ongoing scientific debates whether ancestral genome architecture (proximity to TEs and methylation patterns) or selection following whole genome duplication forms the patterns. Placing the results in the context of observations made in other systems can enrich the biological significance of this study.

Since the manuscript presents assemblies of highly complex polyploid genomes, I recommend employing an additional quality check, looking at the gene coverage distributions of single and multicopy genes from BUSCO set. For example, this can be done using `calculate_AG.py` tool from <https://github.com/shelkmike/Mabs>

Because, for example, a high proportion of BUSCO genes (50%) are still assembled in a single copy in polyploid genomes, which is not necessarily what one might expect if re-diploidization is still at the very early stages and karyotypes are stably doubled or tripled. Maybe it makes sense to do BUSCO analysis for each subgenome separately in this case.

Similar concerns are applied to gene retention patterns between the subgenomes. Additional quality checks for assembly quality are required to check if, for example, (1) the gene is missing from the assembly rather than from the genome or (2) the gene is retained in multicopy due to divergence between two alleles leading to the unmerged assembly of different haplotypes. This analysis requires the calculation of coverage in retained genes: (1) missing gene copy from the assembly will most probably result in a doubled coverage on the other copy, (2) while retention pattern due to separate assemblies of diverged alleles (haplotypes) would result in lower (~half) coverage.

These additional QC checks are important, as polyploid genomes are way more difficult to assemble and gene loss patterns can potentially be artifacts of misassemblies.

For example, Supplementary Fig. 1 shows a high degree of contact between pseudochromosomes in

polyploid assemblies, which could be a sign of misassemblies.

Moreover, expression bias of the C subgenome in all ploidy combinations does not support the observed differential bias in gene retention in tetraploids vs hexaploids. This again, suggests potential issues with the genome assemblies of the polyploids leading to erroneous RNAseq mapping.

Minor:

L268-270 hard to understand what you mean here: that tetraploid species originated independently? That C subgenome in both tetraploids is dominant? That the presence of A subgenome in hexaploids shifted dominance? Please split the sentence and make it more clear what are the conclusions of this subsection.

L29 polyploid nucleus

L83 habitats?

Figure 1 is way too busy, synteny blocks are unreadable

Figures in general are very hard to read, with some missing labels on the bars (Ext Fig 4e, for example). Abbreviations of species names used without ploidy indication are also not very convenient.

L. 113 Extended fig. 1

The part on lines 150-158 about the expected patterns of gene tree inconsistencies under ILS needs a better explanation. A reference to a previous paper using this approach doesn't make it easy for a reader to understand.

Author Rebuttal to Initial comments

Reviewers' Comments:

Reviewer #1:

Remarks to the Author:

The manuscript integrates extensive phylogenomics, comparative transcriptomic analyses to investigate the impact of polyploidization on species diversification using an extraordinary plant system, the polyploid bamboos. The study found that bamboo subgenomes exhibit remarkable karyotype stability, with parallel subgenome dominance in the two tetraploid clades and a gradual shift of dominance in the hexaploid clade. Furthermore, they found that polyploidy and subgenome dominance explain the genetic basis of biological traits of woody bamboos. This study provides valuable insights into the polyploidization and evolutionary history in the remarkable polyploid bamboo system. Here I have some questions about the data interpretation, however, not undermining the overall merit of the study.

Response: Thank you for your positive comment and valuable suggestions. As suggested, we did new analyses to test putative introgression events and homoeologous exchanges, and further

explore the impact of subgenome dominance on the evolution of specific traits in woody bamboos. We integrated these new results, giving additional evidence for our main conclusions, into the thoroughly revised manuscript.

Major concerns:

1) There are several possible reasons for the extensive topological discordance observed among coalescent-based trees. One plausible explanation could be the insufficient inference of introgression events among different genera and species. As the authors did not clarify how they addressed introgression events in their data, it is difficult to assess the extent to which such events may have affected their results.

Response: Thanks for your suggestion and we agree that there are many possible factors underlying the observed conflicting phylogenetic relationships. In the previous manuscript, we focused on incomplete lineage sorting (ILS) and hybridization as possible reasons based on the distributions of individual gene trees supporting different conflicting topologies. For example, the detection of unequal percentage of gene trees supporting the two minor alternative topologies, implying occurrence of ancient hybridization. Following your suggestion, we inferred putative introgression events with the widely used software QuIBL and HyDe. The introgression signal was low (see new Extended Fig. 3d and Supplementary Table 10), mainly surrounding the

A subgenome and the herbaceous bamboo lineage. This was consistent with our previous inferences based on PhyloNet analysis. Taken together, multiple pieces of evidence point to hybridization as the main reason for the extensive topological discordance observed, with subsequent introgression also playing a role. We added results of new analyses in the manuscript and revised the relevant content accordingly.

2) The mosaic chromosome of Chr09D in TWB groups displays a distinct fusion pattern between chr09D and chr2C. However, the authors did not elaborate on this point, leaving

room for further interpretation. One possible explanation is that homoeologous exchange, which is common among polyploid species, may have caused this mix of autopolyploid genome fragments in the allopolyploid genome. It would be interesting to know if the authors have identified smaller fragments showing a similar pattern in other polyploid genomes.

Response: Thanks for your suggestion. We looked more into this interesting mosaic chromosome as well and investigated putative homoeologous exchanges (HEs) in the polyploid genomes. Chr2C and chr9D are not homoeologous chromosomes between subgenomes within the TWB bamboos. The formation of this mosaic chromosome could be considered as a one-way translocation event of a large segment from chr2C to chr9D while no sequence of chr9D was transferred to chr2C. Moreover, the translocation occurred in the most recent common ancestor of all the three species of TWBs; these species all share the translocated segment ranging from 38.9 Mb to 54.8 Mb in size, and the region around the break point is well-conserved in synteny among three genomes. This is clearly demonstrated in the revised Extended Fig. 2d.

Therefore, the chr9D mosaic chromosome could have formed shortly after polyploidization in the common ancestor of TWBs. In addition, we also identified smaller fragments putatively affected by HEs, although it is difficult to analyze in polyploid bamboos due to the lack (probably because of extinction) of extant diploid progenitors, which are usually used as reference in similar analyses, such as in oat¹, peanut² and strawberry³. Based on a phylogenetic approach, we observed putative HEs among subgenomes at a low rate of 0.43% to 1.27% of genes (see new Supplementary Table 11 and Supplementary Fig. 17). This was consistent with and

provided additional evidence for the observed striking karyotype stability of woody bamboos, probably related to their infrequent flowering and reproduction, thus providing less chance of rearrangement during meiosis.

References

1. Peng, Y. *et al.* Reference genome assemblies reveal the origin and evolution of allohexaploid oat. *Nat. Genet.* **54**, 1248–1258 (2022).
2. Bertoli, D. J. *et al.* The genome sequence of segmental allotetraploid peanut *Arachis hypogaea*. *Nat. Genet.* **51**, 877–884 (2019).
3. Edger, P. P. *et al.* Origin and evolution of the octoploid strawberry genome. *Nat. Genet.* **51**, 541–547 (2019).

3) The authors of this study conducted extensive research on polyploid genome evolution. However, their findings regarding the influence of polyploidy on the diversification of specific traits are relatively weak. For instance, it is unclear whether the authors found evidence to support the dominance of the C subgenome in gene family size, novel genes, and PSGs related to woody culms and flowering. Therefore, the authors should provide additional evidence to strengthen their argument regarding the impact of subgenome dominance on the diversification of specific traits.

Response: Thank you for pointing this out and we conducted new analyses to strengthen our argument. In addition to gene family size, new genes, and positively selected genes (PSGs) identified in the previous manuscript, we also investigated specific expressed genes mostly found in the shoot and inflorescence in two representative woody bamboo species (*D. sinicus* and *P. edulis*). Importantly, we took more consideration of subgenomes in relating the genomic changes to the evolution of specific traits in woody bamboos. Among the genomic changes documented, we indeed found that the generally dominant C subgenome played a more important role than other subgenomes. For instance, functional enrichment analyses of expanded gene families from the C subgenome gave more significant results such as the GO terms associated with plant vegetative growth and development, in contrast to no significantly enriched GO term identified for the B subgenome. For the new genes examined, many were found to be specifically expressed in the shoot of *P. edulis* and *D. sinicus* and with a generally higher transcriptome age index (TAI) for the C subgenome. For

the PSGs identified, the C subgenome was much enriched compared to other subgenomes. More importantly, the genes experiencing two or more genomic changes identified above (e.g., expanded gene copies and positive selection) are C- subgenome overrepresented. Furthermore, many important genes involved in the unique life cycle of woody bamboos were all from the C subgenome. The new results are mainly presented in the revised Fig. 5a,b and new Extended Fig. 7c,d. In sum, we provided more evidence for the argument regarding the importance of subgenome dominance on the evolution of specific traits in woody bamboos. This also strengthens the aim of the manuscript.

4) Reproductive research. As the authors have conducted extensive in-depth analyses on genome and subgenome evolution, it would be valuable for the scientific community if they make their analysis pipeline and code publicly available. This would enable other researchers to reproduce and expand upon their work in future studies.

Response: This is an important suggestion. To cope with the challenges in analyses associated with multiple species, we developed some new code and an analysis pipeline. Following your suggestion, we have uploaded the new pipeline and code to GitHub (<https://github.com/yunlongliukm/BGSP>), which would be publicly available after acceptance of the manuscript. We hope that other researchers will find them helpful to carry out similar studies.

Minor:

1) The title. Since the research examined subgenome dominance in the remarkable polyploid bamboo system, it might be better to modify the title to include the “bamboo” instead of only “grasses”.

Response: Thanks for your suggestion and we added “bamboo” into the title. The revised title is **“Diversification induced by dynamic subgenome dominance in bamboos, the**

world's largest grasses".

2)Line 142. "...the set of 456 genes 26". It is unclear for the meaning of 2 numbers.

Response: We identified 456 "perfect-copy" syntenic genes for analyses of subgenome identification, however, 26 of these genes have gene trees poorly resolved or putatively affected by gene conversion. We thus excluded these 26 outlier genes from the data set for subsequent analyses. We revised this sentence to make it clearer.

3)Line 194. "6 and 11 in the C subgenome" – unclear use of the numbers.

Response: The two numbers represent chr6 and chr11 and we revised this sentence to make it clear.

4)Line 381-383. How many PSGs are from C subgenomes. Do they show any enrichment for C subgenome?

Response: Most of the PSGs identified are from the C subgenome, such as in *P. edulis* which is 146:46 for C:D subgenomes and they did show enrichment for the C subgenome. We revised the related sentence in the manuscript to strengthen the importance of the dominant subgenome.

Reviewer #2:

Remarks to the Author:

The submitted manuscript represents a great effort in genome assembly and analysis of subgenome dominance in the bamboo genus. This potentially very strong study has several issues. First, it is very hard to digest for a reviewer and future general reader. The authors need to think about how to present the data and results in a lighter way. Second, I feel that initial quality checks of genome assemblies are lacking, which may affect the rest of the results.

Response: Thanks for your comments and critical suggestions. First, we critically revised the

manuscript to make it more focused on the analysis of subgenome dominance and streamlined the results, including removal of certain results (e.g., previous Fig. 4a) from the main figures and some content not closely related to the focus from the main text. We also made substantial revisions of some figures (Figs. 1 and 3-5) and presented more interpretation of results closely related to the aim of our manuscript. Second, we did additional quality checks of the genome assemblies following your suggestions, and obtained positive results as detailed below. This gives us more confidence in our genome assemblies and the suggested quality checks did not affect the main results.

The topic of subgenome dominance is very interesting with ongoing scientific debates whether ancestral genome architecture (proximity to TEs and methylation patterns) or selection following whole genome duplication forms the patterns. Placing the results in the context of observations made in other systems can enrich the biological significance of this study.

Response: Thank you for your suggestion. We revised and added discussion of our results in the context of observations made in other polyploid systems in the revised manuscript but limited to a few sentences due to the limited word count of the journal (ca. 4,000 words). We also found that ancestral genome architecture (TE density and methylation patterns) could determine the formation of subgenome dominance. Our data also provide new insights into the evolution of subgenome dominance, particularly noteworthy is the possible shift of subgenome dominance after addition of a third subgenome in the hexaploid bamboos.

Since the manuscript presents assemblies of highly complex polyploid genomes, I recommend employing an additional quality check, looking at the gene coverage distributions of single and multicopy genes from BUSCO set. For example, this can be done using calculate_AG.py tool from <https://github.com/shelkmike/Mabs>.

Because, for example, a high proportion of BUSCO genes (50%) are still assembled in a single copy in polyploid genomes, which is not necessarily what one might expect if re-diploidization is still at the very early stages and karyotypes are stably doubled or tripled. Maybe it makes

sense to do BUSCO analysis for each subgenome separately in this case.

Response: Many thanks for your recommendation of this recently developed software, which is an efficient tool for evaluating the quality of polyploid genome assemblies.

And we agree that examining the gene coverage distributions of single- and multi- copy genes is an important approach. Using the calculate_AG.py tool, we obtained high AG (Accurately assembled Genes) scores for all the nine polyploid genome assemblies (1,520 to 2,155, see new Supplementary Fig. 3), mostly contributed by the true multicopy orthogroups. This is in sharp contrast to that observed in the two diploid genome assemblies with AG scores mostly contributed by the single-copy orthogroups. More importantly, we also found consistent coverage distributions for single- and multi-copy genes from the BUSCO set (see new Extended Fig. 1n), suggesting the accuracy of assembly for single- and multi-copy genes. For the question about ~50% of BUSCO genes assembled in a single copy, we re-analyzed the data and obtained the identical result. And the analysis at the subgenome level revealed 63.3%-80.1% and 49.3%-65.3% BUSCO complete scores for individual subgenomes in the tetraploid and hexaploid genomes (results not shown in the manuscript due to the limited word count), respectively. This could be explained by the distinctive evolutionary histories of the bamboo genomes. Although the karyotypes are generally stable in woody bamboos, the massive loss of genes could still be expected considering the timing of polyploidization, which occurred about 20 million years ago. The loss of genes and the evolution of genomic rearrangements may be decoupled, and the slow evolution of woody bamboo karyotypes indeed differs from some other recent polyploids such as maize¹ and oat^{2,3}. For instance, the polyploidization of oat was just ~0.5 million years ago but it has a mosaic genome.

The slow evolution of karyotypes in woody bamboos might be related to their unique life cycle (infrequent flowering, 30 to 60 years), with less chance of genomic rearrangements during

meiosis. The slow evolution is also supported by the comparison of karyotypes between herbaceous and woody bamboos, with the later more resembling the ancestral grass karyotype despite experiencing polyploidizations. In this case, we tend to consider that BUSCO analysis at the subgenome level may be not appropriate in evaluating the quality of ancient polyploid genome assemblies here.

References

1. Schnable, P. S. *et al.* The B73 maize genome: complexity, diversity, and dynamics. *Science* **326**, 1112-5 (2009).
2. Kamal, N. *et al.* The mosaic oat genome gives insights into a uniquely healthy cereal crop. *Nature* **606**, 113-119 (2022).
3. Peng, Y. *et al.* Reference genome assemblies reveal the origin and evolution of allohexaploid oat. *Nature Genetics* **54**, 1248-1258 (2022).

Similar concerns are applied to gene retention patterns between the subgenomes. Additional quality checks for assembly quality are required to check if, for example, (1) the gene is missing from the assembly rather than from the genome or (2) the gene is retained in multicopy due to divergence between two alleles leading to the unmerged assembly of different haplotypes. This analysis requires the calculation of coverage in retained genes: (1) missing gene copy from the assembly will most probably result in a doubled coverage on the other copy, (2) while retention pattern due to separate assemblies of diverged alleles (haplotypes) would result in lower (~half) coverage. These additional QC checks are important, as polyploid genomes are way more difficult to assemble and gene loss patterns can potentially be artifacts of misassemblies. For example, Supplementary Fig. 1 shows a high degree of contact between pseudochromosomes in polyploid assemblies, which could be a sign of misassemblies. Moreover, expression bias of the C subgenome in all ploidy combinations does not support the observed differential bias in gene retention in tetraploids vs hexaploids. This again, suggests potential issues with the genome assemblies of the polyploids leading to erroneous RNAseq mapping.

Response: As suggested, we paid further attention to the quality of the genome assemblies and did additional coverage analyses for all the single- and multi-copy genes within the polyploid genomes using short sequencing reads. Once again, we obtained normal and consistent coverage

distributions for the single-copy (loss in one subgenome) and two-copies (retained in both subgenomes) genes in all the six tetraploid genomes, as well as for the single-copy (loss in two subgenomes), two-copy (loss in one subgenome) and three-copy (retained in all three subgenomes) genes in all the three hexaploid genomes (see new Supplementary Fig. 19). These results support the observed patterns of gene fractionation, which is the same for a certain subgenome among different species within the same clade. If genes were extensively misassembled, these misassembled genes could be randomly distributed among subgenomes and we would not obtain a consistent pattern of gene fractionation for the same subgenome from different species. Although some noise may exist in the Hi-C map, we sequenced multiple species for each of three polyploid clades and they were *de novo* assembled. Their genomes exhibit highly conserved synteny, such as the shared fused chr9D in three temperate woody bamboos; we would not likely observe the same mistake three times in obtaining the results. The differential bias in tetraploids vs. hexaploids can be explained by the different polyploidization events of tetraploids and hexaploids. The A subgenome was added to the hexaploids much later than the B and C subgenomes, which may explain why it had more genes retained. Moreover, the expression bias within the hexaploids is dynamic and differs among species, with the bias generally toward to the C subgenome in the early-diverging species shifting to the A subgenome in the later-diverging species and it varies among different tissues. In all, after finishing all the quality analyses suggested, we are more confident regarding the quality of our genome assemblies. Nevertheless, possibility of mis-assemblies cannot be completely ruled out, even though all analyses suggested that it was not a problem here and would not affect the rest of the results.

Subsequently, we revised the relevant paragraph of the results.

Minor:

L268-270 hard to understand what you mean here: that tetraploid species originated independently? That C subgenome in both tetraploids is dominant? That the presence of A subgenome in hexaploids shifted dominance? Please split the sentence and make it more clear what are the conclusions of this subsection.

Response: We revised the text to make these points clearer. We sampled two tetraploid clades of bamboos, each with three species. And phylogenomic analyses supported their independent origins with different combinations of subgenomes (i.e., BBCC for neotropical woody bamboo clade and CCDD for temperate woody bamboo clade). The hexaploid clade of bamboo (AABBCC) was formed with the neotropical bamboo as one parent. The C subgenome was found to be dominant consistently across all three species in both tetraploid clades and this occurred in parallel. In the hexaploids, the pattern of dominance was more complex. At the sequence level of gene fractionation, the A subgenome appears to be dominant in all three species; however, for gene density and the proportion of genes expressed it is still the C subgenome that is dominant. At the expression level, we found that the C subgenome appeared to be the dominant one in the early-diverging species of the hexaploid clade but in the other two species the A subgenome was more biased. In all, the subgenome dominance at the expression level is more dynamic in the hexaploid evolution.

L29 polyploid nucleus Response:

Corrected. L83 habitats?

Response: We revised this sentence and corrected it.

Figure 1 is way too busy, synteny blocks are unreadable. Figures in general are very hard to read, with some missing labels on the bars (Ext Fig 4e, for example).

Abbreviations of species names used without ploidy indication are also not very convenient.

Response: Thanks. We critically revised the main figures to make them clearer and easier to follow. We also corrected the missing labels as you pointed out and checked all of figures to avoid similar problems. We added ploidal levels in the appropriate places to make the results of related figures and tables more understandable as suggested.

L. 113 Extended fig. 1 Response:

Corrected.

The part on lines 150-158 about the expected patterns of gene tree inconsistencies under ILS needs a better explanation. A reference to a previous paper using this approach doesn't make it easy for a reader to understand.

Response: Thank you for noting this. Both incomplete lineage sorting (ILS) and hybridization can lead to topological inconsistencies of gene trees. When ILS occurs alone, the frequencies of two minor alternative topologies are expected to be equal, while in the presence of hybridization, one of the two minor topologies would be recovered more frequently than the other. We have revised this sentence to make it clearer.

Decision Letter, first revision:

3rd Oct 2023

Dear Dr. Li,

Thank you for submitting your revised manuscript "Diversification induced by dynamic subgenome dominance in bamboos, the world's largest grasses" (NG-A62141R). It has now been seen by the original referees and their comments are below. The reviewers find that the paper has improved in revision, and therefore we'll be happy in principle to publish it in Nature Genetics, pending minor revisions to satisfy the referees' final requests and to comply with our editorial and formatting guidelines.

Sincerely,
Wei

Wei Li, PhD
Senior Editor
Nature Genetics
New York, NY 10004, USA
www.nature.com/ng

Reviewer #1 (Remarks to the Author):

After reviewing the authors' responses to the reviewer's comments and the revisions made to the manuscript, I believe as a reviewer that the authors have adequately addressed and resolved the issues and suggestions I raised. The authors conducted additional analyses and discussions on the questions I raised in the first review, including possible gene introgression, homoeologous recombination, and the influence of the dominant subgenome on the evolution of specific traits. These new results provide further support for the main conclusions of the paper. The authors also revised the title and uploaded the analysis pipeline and code as I suggested. With the revisions made, the scientific merit and innovativeness of the paper are further enhanced, providing essential insights into the origin and evolution of plant polyploids.

Author Rebuttal, first revision:

Response to the referees

(Nature Genetics submission NG-A62141R)

Reviewer #1:

Remarks to the Author:

After reviewing the authors' responses to the reviewer's comments and the revisions made to the manuscript, I believe as a reviewer that the authors have adequately addressed and resolved the issues and suggestions I raised. The authors conducted additional analyses and discussions on the questions I raised in the first review, including possible gene introgression, homoeologous recombination, and the influence of the dominant subgenome on the evolution of specific traits. These new results provide further support

for the main conclusions of the paper. The authors also revised the title and uploaded the analysis pipeline and code as I suggested. With the revisions made, the scientific merit and innovativeness of the paper are further enhanced, providing essential insights into the origin and evolution of plant polyploids.

Response: Thank you for your positive comments.

Reviewer #2:

None

Response: Thanks.

Final Decision Letter:

8th Feb 2024

Dear Dr. Li,

I am delighted to say that your manuscript "Genome assemblies of 11 bamboo species highlight diversification induced by dynamic subgenome dominance" has been accepted for publication in an upcoming issue of Nature Genetics.

Your paper will be published online after we receive your corrections and will appear in print in the next available issue. You can find out your date of online publication by contacting the Nature Press Office (press@nature.com) after sending your e-proof corrections.

Please note that *Nature Genetics* is a Transformative Journal (TJ). Authors may publish their research with us through the traditional subscription access route or make their paper immediately open access through payment of an article-processing charge (APC). Authors will not be required to make a final decision about access to their article until it has been accepted. Find out more about Transformative Journals

Authors may need to take specific actions to achieve compliance with funder and institutional open access mandates. If your research is supported by a funder that requires immediate open access (e.g. according to Plan S principles) then you should select the gold OA route, and we will direct you to the compliant route where possible. For authors selecting the subscription publication route, the journal's standard licensing terms will need to be accepted, including <https://www.nature.com/nature-portfolio/editorial-policies/self-archiving-and-license-to-publish>. Those licensing terms will supersede any other terms that the author or any third party may assert apply to any version of the manuscript.

If you have not already done so, we invite you to upload the step-by-step protocols used in this manuscript to the Protocols Exchange, part of our on-line web resource, natureprotocols.com. If you complete the upload by the time you receive your manuscript proofs, we can insert links in your article that lead directly to the protocol details. Your protocol will be made freely available upon publication of your paper. By participating in natureprotocols.com, you are enabling researchers to more readily reproduce or adapt the methodology you use. [Natureprotocols.com](http://natureprotocols.com) is fully searchable, providing your protocols and paper with increased utility and visibility. Please submit your protocol to <https://protocolexchange.researchsquare.com/>. After entering your nature.com username and password you will need to enter your manuscript number (NG-A62141R1). Further information can be found at <https://www.nature.com/nature-portfolio/editorial-policies/reporting-standards#protocols>

Sincerely,

Wei Li, PhD
Senior Editor
Nature Genetics
New York, NY 10004, USA
www.nature.com/ng